# Impairments in laterodorsal tegmentum to VTA projections underlie glucocorticoid-triggered reward deficits

Bárbara Coimbra[1,2], Carina Soares-Cunha[1,2], Sónia Borges[1,2], Nivaldo AP Vasconcelos[1,2], Nuno Sousa[1,2]*, Ana João Rodrigues[1,2]*

[1]Life and Health Sciences Research Institute (ICVS), School of Medicine, University of Minho, Braga, Portugal; [2]ICVS/3B's–PT Government Associate Laboratory, Braga/Guimarães, Portugal

**Abstract** Ventral tegmental area (VTA) activity is critical for reward/reinforcement and is tightly modulated by the laterodorsal tegmentum (LDT). *In utero* exposure to glucocorticoids (iuGC) triggers prominent motivation deficits but nothing is known about the impact of this exposure in the LDT-VTA circuit. We show that iuGC-rats have long-lasting changes in cholinergic markers in the LDT, together with a decrease in LDT basal neuronal activity. Interestingly, upon LDT stimulation, iuGC animals present a decrease in the magnitude of excitation and an increase in VTA inhibition, as a result of a shift in the type of cells that respond to the stimulus. In agreement with LDT-VTA dysfunction, we show that iuGC animals present motivational deficits that are rescued by selective optogenetic activation of this pathway. Importantly, we also show that LDT-VTA optogenetic stimulation is reinforcing, and that iuGC animals are more susceptible to the reinforcing properties of LDT-VTA stimulation.

DOI: https://doi.org/10.7554/eLife.25843.001

*For correspondence:
njcsousa@med.uminho.pt (NS);
ajrodrigues@med.uminho.pt (AJR)

Competing interests: The authors declare that no competing interests exist.

## Introduction

The ventral tegmental area (VTA) is an heterogeneous brain region containing distinctive neuronal populations essential for the expression of motivated behaviors and reinforcement (*Berridge and Robinson, 1998*; *Wise, 2004*; *Bayer and Glimcher, 2005*; *Fields et al., 2007*; *Berridge, 2007*; *van Zessen et al., 2012*). The VTA comprises dopaminergic (~65%), GABAergic (~30%), and glutamatergic neurons (~5%) (*Nair-Roberts et al., 2008*; *Yamaguchi et al., 2011*) that receive inputs from diverse brain regions, including the laterodorsal tegmentum (LDT) (*Woolf and Butcher, 1986*; *Cornwall et al., 1990*; *Oakman et al., 1995*; *Oakman et al., 1999*).

Several studies have shown that exposure to unexpected rewards, or cues that predict rewards, can activate VTA dopaminergic neurons culminating in the release of dopamine in the nucleus accumbens (NAc) (*Roitman et al., 2004*; *Stuber et al., 2005*; *Stuber et al., 2008*; *Schultz et al., 1997*; *Bromberg-Martin et al., 2010*). Importantly, this activity is tightly modulated by cholinergic projections (*Omelchenko and Sesack, 2005*; *Omelchenko and Sesack, 2006*), with an additional contribution of glutamatergic projections, arising from the LDT (*Cornwall et al., 1990*; *Oakman et al., 1999*; *Lammel et al., 2012*). This input is vital for the activity of dopaminergic cells in the VTA, facilitating dopamine-related behaviors involved in reward signaling or encoding reward prediction signals (*Lodge and Grace, 2006*). In agreement, recent studies have shown that optogenetic stimulation of LDT neurons that project to the VTA enhances conditioned place preference (*Lammel et al., 2012*) and operant responses in rodents (*Steidl and Veverka, 2015*).

Notably, different labs have shown that the mesolimbic system is particularly vulnerable to the effects of prenatal stress/high levels of glucocorticoids (GCs) (*Matthews, 2000*; *Boksa and El-*

*Khodor, 2003*; *McArthur et al., 2005*; *Leão et al., 2007*; *Rodrigues et al., 2011*; *Borges et al., 2013a*; *Soares-Cunha et al., 2014*). These changes may increase the risk to develop different neuro-psychiatric disorders in adulthood, namely depression, anxiety and addiction (*Seckl, 2008*; *Rodrigues et al., 2012*). Surprisingly, very few studies have focused on the impact of stress/GCs in the cholinergic system. This is particularly intriguing because GCs can induce acetylcholine release (*Finkelstein et al., 1985*; *Gilad et al., 1985*; *Imperato et al., 1989*) and bind to GC-responsive elements of cholinergic enzymes, namely choline acetyltransferase (ChAT) and acetylcholine esterase (AChE) to control their expression (*Berse and Blusztajn, 1997*). In accordance, we have previously shown that prenatal GC exposure induces a long-lasting hyperanxious state associated with an increase in the recruitment of cholinergic cells from the LDT (*Borges et al., 2013b*), suggesting that GCs are able to *program* the LDT, which prompted us to evaluate the impact of prenatal GC in the LDT-VTA circuitry and its impact in reward-related behaviors.

## Results

### Sustained cholinergic dysfunction in iuGC animals

Previous data from our team suggested that LDT cholinergic cells were differentially recruited in response to an adverse stimulus (*Borges et al., 2013b*) in a model of *in utero* GC (iuGC) exposure at gestation days 18 and 19 (*Blaha and Winn, 1993*). Considering this, we first evaluated the impact of GCs on the cholinergic circuitry of iuGC animals. We quantified ChAT$^+$ cells in the LDT of 3, 30 and 90 days old animals (*Figure 1a–c*) and observed an effect of iuGC treatment (Two-way ANOVA; $F_{(1,25)}$ = 19.31, p=0.0002). iuGC animals had a significant increase in the density of the cholinergic population of the LDT at 30 days of age (*post-hoc* Bonferroni; CTR$_{(30 \text{ days})}$ vs. iuGC$_{(30 \text{ days})}$: $t_{(25)}$ = 2.616, p=0.0446) that persisted until adulthood (*post-hoc* Bonferroni; CTR$_{(90 \text{ days})}$ vs. iuGC$_{(90 \text{ days})}$: $t_{(25)}$ = 3.971, p=0.0016). Other brain regions containing cholinergic neurons such as the nucleus basalis of Meynert or the NAc remained unaltered (*Figure 1—figure supplement 1*).

We next evaluated gene and protein expression levels of ChAT and AChE in the LDT (*Figure 1d–h*). We found a significant effect of iuGC treatment in ChAT (Two-way ANOVA; $F_{(1,24)}$ = 26.27, p<0.0001) and AChE (Two-way ANOVA; $F_{(1,23)}$ = 15.71, p=0.0006) gene expression. ChAT gene expression levels were increased at 3 and 30 days of age in iuGC animals (*Figure 1d*; *post-hoc* Bonferroni; CTR$_{(3 \text{ days})}$ vs. iuGC$_{(3 \text{ days})}$: $t_{(24)}$ = 3.383, p=0.0074; CTR$_{(30 \text{ days})}$ vs. iuGC$_{(30 \text{ days})}$: $t_{(24)}$ = 3.053, p=0.0164; CTR$_{(90 \text{ days})}$ vs. iuGC$_{(90 \text{ days})}$: $t_{(24)}$ = 2.494, p=0.059). Decreased AChE levels were found in adult iuGC animals (*Figure 1e*; *post-hoc* Bonferroni; CTR$_{(3 \text{ days})}$ vs. iuGC$_{(3 \text{ days})}$: $t_{(23)}$ = 1.24, p=0.6827; CTR$_{(30 \text{ days})}$ vs. iuGC$_{(30 \text{ days})}$: $t_{(23)}$ = 2.244, p=0.1043; CTR$_{(90 \text{ days})}$ vs. iuGC$_{(90 \text{ days})}$: $t_{(23)}$ = 3.298, p=0.0094). We also evaluated the levels of another cholinergic marker, the vesicular acetylcholine transporter (VAChT) and found that mRNA levels were unchanged between groups in the LDT (*Figure 1—figure supplement 2*).

Two-way ANOVA showed a significant effect of iuGC treatment in ChAT ($F_{(1,23)}$ = 32.82, p<0.0001) and AChE protein expression ($F_{(1,18)}$ = 425.08, p<0.0001). Western blot analysis confirmed the upregulation of ChAT (*Figure 1f–g*; *post-hoc* Bonferroni; CTR$_{(3 \text{ days})}$ vs. iuGC$_{(3 \text{ days})}$: $t_{(23)}$ = 4.401, p=0.0006; CTR$_{(30 \text{ days})}$ vs. iuGC$_{(30 \text{ days})}$: $t_{(23)}$ = 2.762, p=0.0333; CTR$_{(90 \text{ days})}$ vs. iuGC$_{(90 \text{ days})}$: $t_{(23)}$ = 2.712, p=0.0373); and downregulation of AChE in the iuGC group at all ages tested (*Figure 1f and h*; *post-hoc* Bonferroni; CTR$_{(3 \text{ days})}$ vs. iuGC$_{(3 \text{ days})}$: $t_{(18)}$ = 4.73, p=0.0005; CTR$_{(30 \text{ days})}$ vs. iuGC$_{(30 \text{ days})}$: $t_{(18)}$ = 3.157, p=0.0164; *post-hoc* Bonferroni CTR$_{(90 \text{ days})}$ vs. iuGC$_{(90 \text{ days})}$: $t_{(18)}$ = 3.349, p=0.0107).

Considering the heterogeneous nature of LDT inputs to the VTA, we also assessed the impact of iuGC exposure on glutamatergic and GABAergic markers (*Figure 1—figure supplement 3a–c,e–g*). Gene and protein expression levels of glutamate transporter EAAC1 and GAD1/67 + GAD2/65 were not significantly affected by iuGC exposure.

We also decided to evaluate the expression levels of glucocorticoid receptor (GR) since early life adversity has been shown to change GR epigenetic status. We found no differences between groups regarding GR expression (*Figure 1—figure supplement 3d,h*).

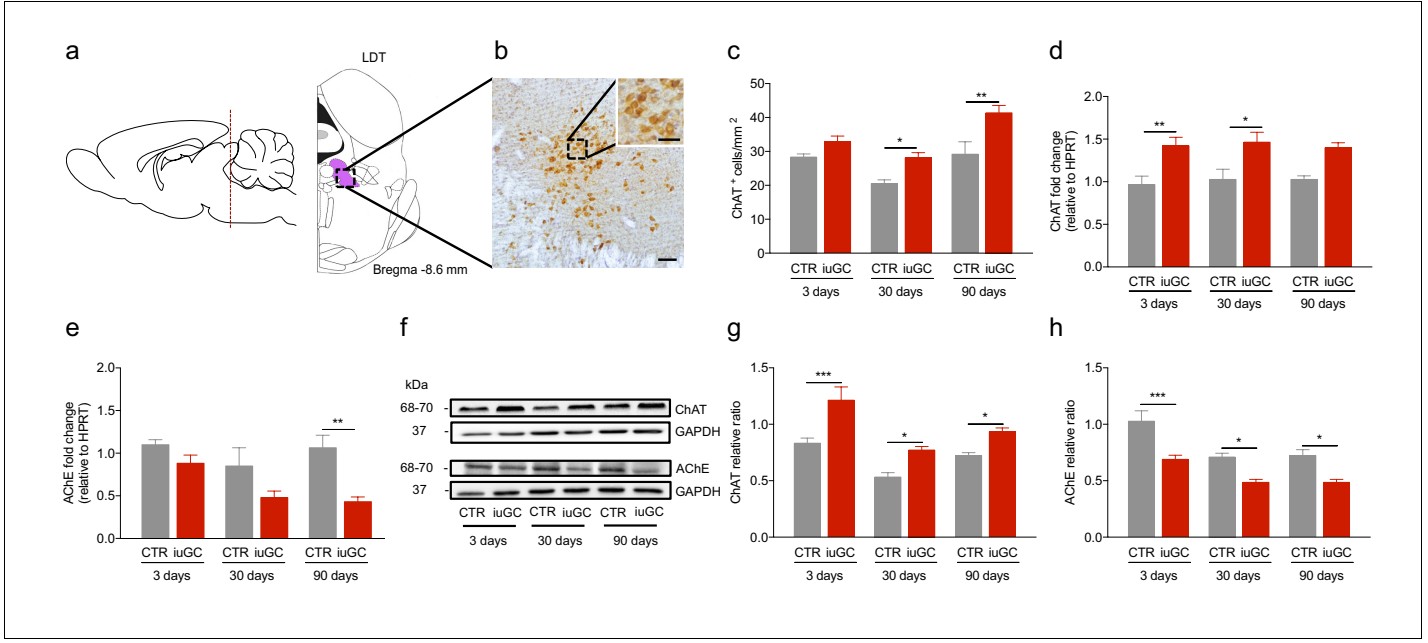

**Figure 1.** Prenatal exposure to glucocorticoids alters LDT cholinergic system. (**a**) Schematic representation of the LDT. (**b**) Coronal section of the LDT showing ChAT immunohistochemistry. (**c**) iuGC animals present increased number of ChAT[+] cells in the LDT at postnatal day 30 and 90. (**d**) Real-time PCR analysis revealed that ChAT mRNA levels are increased in the LDT of iuGC animals from postnatal day 3. (**e**) Conversely, AChE mRNA levels are decreased at postnatal day 90 ($n_{CTR}$ = 4; $n_{iuGC}$ = 5). (**f**) Representative immunoblot of ChAT and AChE in the LDT of 3, 30 and 90 days old animals. (**g**) Protein quantification confirmed the upregulation of ChAT and (**h**) downregulation of AChE in the LDT from postnatal day 3 until adulthood ($n_{CTR}$ = 4; $n_{iuGC}$ = 5). Data represented as mean ±s.e.m. *p<0.05, **p<0.001, ***p<0.0001. Scale bars in b: 100 μm and inset - 50 μm. Additional data is depicted in *Figure 1—figure supplements 1, 2* and *3*.

DOI: https://doi.org/10.7554/eLife.25843.002

The following figure supplements are available for figure 1:

**Figure supplement 1.** iuGC animals do not present changes in the number of cholinergic cells in other regions.

DOI: https://doi.org/10.7554/eLife.25843.003

**Figure supplement 2.** iuGC exposure does not change the expression levels of VAChT.

DOI: https://doi.org/10.7554/eLife.25843.004

**Figure supplement 3.** iuGC exposure does not change the expression levels of GABAergic and glutamatergic markers in the LDT.

DOI: https://doi.org/10.7554/eLife.25843.005

## iuGC treatment impairs the LDT-VTA circuitry

Previous work from our group showed that iuGC animals presented a VTA-NAc hypodopaminergic state (*Leão et al., 2007*; *Borges et al., 2013a*; *Soares-Cunha et al., 2014*). Since the LDT innervates the VTA and can influence NAc dopamine release (*Blaha and Winn, 1993*; *Blaha et al., 1996*; *Forster and Blaha, 2000*; *Forster et al., 2002*; *Miller et al., 2002*; *Forster and Blaha, 2003*), we decided to characterize the LDT-VTA circuit using *in vivo* single cell electrophysiology in anesthetized animals (*Figure 2*).

iuGC treatment significantly decreased the spontaneous activity of LDT neurons (*Figure 2a–b*; $t_{(128)}$ = 4.674, p<0.0001, total number of cells: CTR = 36, iuGC = 94; $n_{CTR}$ = 8, $n_{iuGC}$ = 11 animals). No differences were found in the basal activity of VTA (*Figure 2c–d*; $t_{(70)}$ = 0.0576, p=0.9542, total number of cells: CTR = 40, iuGC = 32; animals: CTR = 8, iuGC = 11).

Electrical stimulation at 0.5 Hz of the LDT evoked both excitatory and inhibitory responses in VTA neurons in contrasting percentages (*Figure 2e*; CTR: 50% excitatory and 30% inhibitory; iuGC: 31% excitatory and 44% inhibitory). We also used defined criteria based on extracellular waveforms and firing rate (*Ungless et al., 2004*; *Ungless and Grace, 2012*; *Totah et al., 2013*) to divide recorded cells into putative DAergic (pDAergic) or GABAergic (pGABAergic) neurons (*Figure 2f–h*). Cells that did not match these criteria were considered to be 'other' types of neurons.

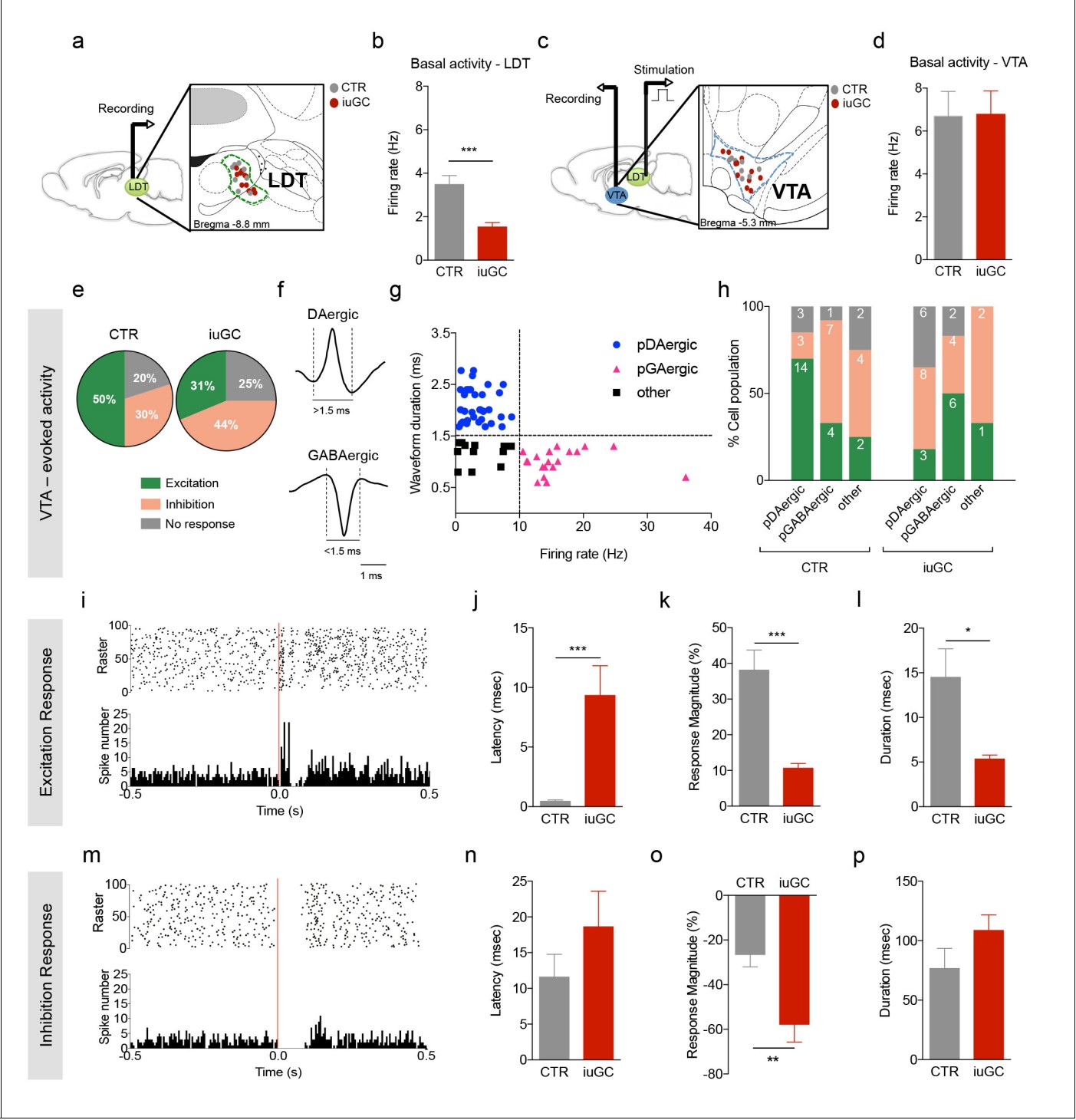

**Figure 2.** Distinct VTA neuronal response to LDT electrical stimulation in iuGC animals. (**a**) Schematic representation of the *in vivo* single-cell electrophysiological recording experiments and electrode placement in the LDT in anesthetized animals. (**b**) iuGC animals present decreased basal activity of LDT neurons in comparison to CTR ($n_{LDT-CTR}$ = 36 cells; $n_{LDT-iuGC}$ = 94 cells). (**c**) Schematic representation of the *in vivo* single-cell electrophysiological recording experiments in the VTA with electric stimulation performed in the LDT; and recording electrode placement. (**d**) The basal activity of the VTA is similar between groups ($n_{VTA-CTR}$ = 40 cells; $n_{VTA-iuGC}$=32 cells). (**e**) Electrical stimulation of the LDT (0.5 Hz) induces excitatory and inhibitory responses in VTA neurons. Pie plots represent the percentage of excitatory, inhibitory and no responses of VTA neurons. (**e–g**) In CTR animals, 50% of neurons present an excitatory response (70% pDAergic, 20% pGABAergic) and 30% present an inhibitory response (21% pDAergic, 50% pGABAergic). iuGC group shows a different profile, with 31% of recorded cells presenting an excitatory response (30% pDAergic, 60% pGABAergic) versus 44% with inhibitory response (57% pDAergic, 29% pGABAergic). (**f**) Representative examples of rat VTA pDAergic and pGABAergic neuronal

*Figure 2 continued on next page*

*Figure 2 continued*

waveforms. (g) Firing rate and waveform duration were used to classify single units into 3 types of neurons. (h) Percentage of each putative neuronal population presenting excitation, inhibition or with no response to LDT stimulation. There is a shift in the percentage of putative DAergic and GABAergic neurons presenting excitatory and inhibitory responses. Numbers in bars represent number of cells in each category. (i, m) Peristimulus time histograms (PSTHs) show LDT-evoked responses of VTA dopamine neurons; (i) excitation; (m) inhibition. (j) VTA neurons that display an excitatory profile in response to LDT electrical stimulus present increased latency to fire in iuGC animals. (k) The magnitude and (l) duration of response of VTA neurons is reduced in iuGC animals. (n) VTA neurons that display an inhibitory response in response to LDT electrical stimulus do not show differences in the latency to fire in both groups. (o) The magnitude of response of inhibited neurons of the VTA is increased in iuGC animals, with no differences in (p) the duration of inhibition in VTA neurons upon LDT stimulation. pDAergic: putative dopaminergic neurons; pGABAergic: putative GABAergic neurons. Data is represented as mean ±s.e.m. *p<0.05, **p<0.001, ***p<0.0001.

DOI: https://doi.org/10.7554/eLife.25843.006

Interestingly, iuGC animals present a shift in the type of neurons that present excitatory responses, with an increase in pGABAergic neurons and concomitant decrease in pDAergic neurons. Excitatory responses were observed in 70% of pDAergic and 20% of pGABAergic neurons in CTR animals versus 30% of pDAergic and 60% of pGABAergic neurons in iuGC animals.

Regarding inhibition, the iuGC group presented an increase in pDAergic neurons together with a decrease in pGABAergic neurons (*Figure 2h*). Briefly, inhibitory responses were observed in 21% of pDAergic and 50% of pGABAergic neurons in CTR animals versus 57% of pDAergic and 28% of pGABAergic neurons in iuGC animals.

The onset of excitation was significantly increased in iuGC group in comparison to control group (*Figure 2j*; CTR: 0.45 ± 0.114 ms vs. iuGC: 9.33 ± 2.49 ms; $t_{(30)}$ = 4.637, p<0.0001). Interestingly, iuGC treatment reduced the magnitude and duration of the excitatory response (*Figure 2k–l*; magnitude: $t_{(30)}$ = 3.723, p=0.0008; duration: $t_{(30)}$ = 2.196, p=0.0360).

LDT stimulation did not affect the latency of inhibition, although there was a trend for increased latency in iuGC animals (*Figure 2n*; CTR: 11.58 ± 3.175 ms vs. iuGC: 18.61 ± 4.979 ms; $t_{(28)}$ = 1.057, p=0.2997). Also, the duration of response was not affected (*Figure 2p*; $t_{(28)}$ = 1.510, p=0.1422). However, the response magnitude of inhibitory responses in the VTA evoked by LDT electrical stimulation was significantly higher in iuGC animals (*Figure 2o*; $t_{(28)}$ = 2.905, p=0.0071).

Altogether, this data demonstrated an imbalance in the excitatory and inhibitory inputs to the VTA when electrically stimulating the LDT.

## Optogenetic activation of LDT terminals in the VTA elicits distinct responses in control and iuGC animals

We next used a combined viral approach to specifically modulate LDT direct inputs to the VTA and exclude the effects of indirect activation of other regions to where LDT projects to. We decided to activate all types of LDT-VTA inputs (and not only cholinergic) because we observed an effect of iuGC exposure in both excitatory and inhibitory VTA responses elicited by LDT activation.

To do so, we injected a viral vector containing a WGA–Cre fusion construct (AAV5–EF1a–WGA–Cre–mCherry) in the VTA, and a cre-dependent ChR2 vector in the LDT (AAV5-EF1a-DIO-hChR2-eYFP). The WGA–Cre fusion protein is retrogradely transported (*Gradinaru et al., 2010*), inducing the expression of cre-dependent ChR2-YFP only in LDT neurons that directly project to the VTA (*Figure 3a–c*). Four weeks post-injection, we observed YFP staining in axonal terminals of LDT neurons in the VTA (*Figure 3b*) and in cell bodies in the LDT (*Figure 3c*).

We performed *in vivo* single cell electrophysiology in the VTA while stimulating LDT terminals in this region, in order to activate the LDT-VTA circuit specifically. As depicted in *Figure 3d*, optogenetic stimulation of LDT terminals (30 pulses of 15 ms at 20 Hz) induced an increase in firing rate of VTA neurons (*Figure 3d*, *post-hoc* Bonferroni test CTR-ChR2: baseline vs. stimulus - $t_{(118)}$ = 5.883, p<0.0001, stimulus vs. post-stimulus - $t_{(118)}$ = 3.749, p=0.0008; iuGC-ChR2: baseline vs. stimulus - $t_{(118)}$ = 10.99, p<0.0001; stimulus vs. post-stimulus - $t_{(118)}$ = 10.88, p<0.0001), eliciting a response in 69% and 71% of recorded cells of CTR and iuGC animals, respectively (*Figure 3e*). Optic stimulation did not affect the latency of stimulus response for the groups (*Figure 3—figure supplement 1*).

In CTR animals, 49% of cells presented increased firing rate upon stimulation, and of these, 69% were pDAergic, 17% pGABAergic and 14% were categorized as 'other' neuronal subtypes. Moreover, 87% of cells that presented inhibitory responses were pGABAergic neurons (*Figure 3f–g*). In

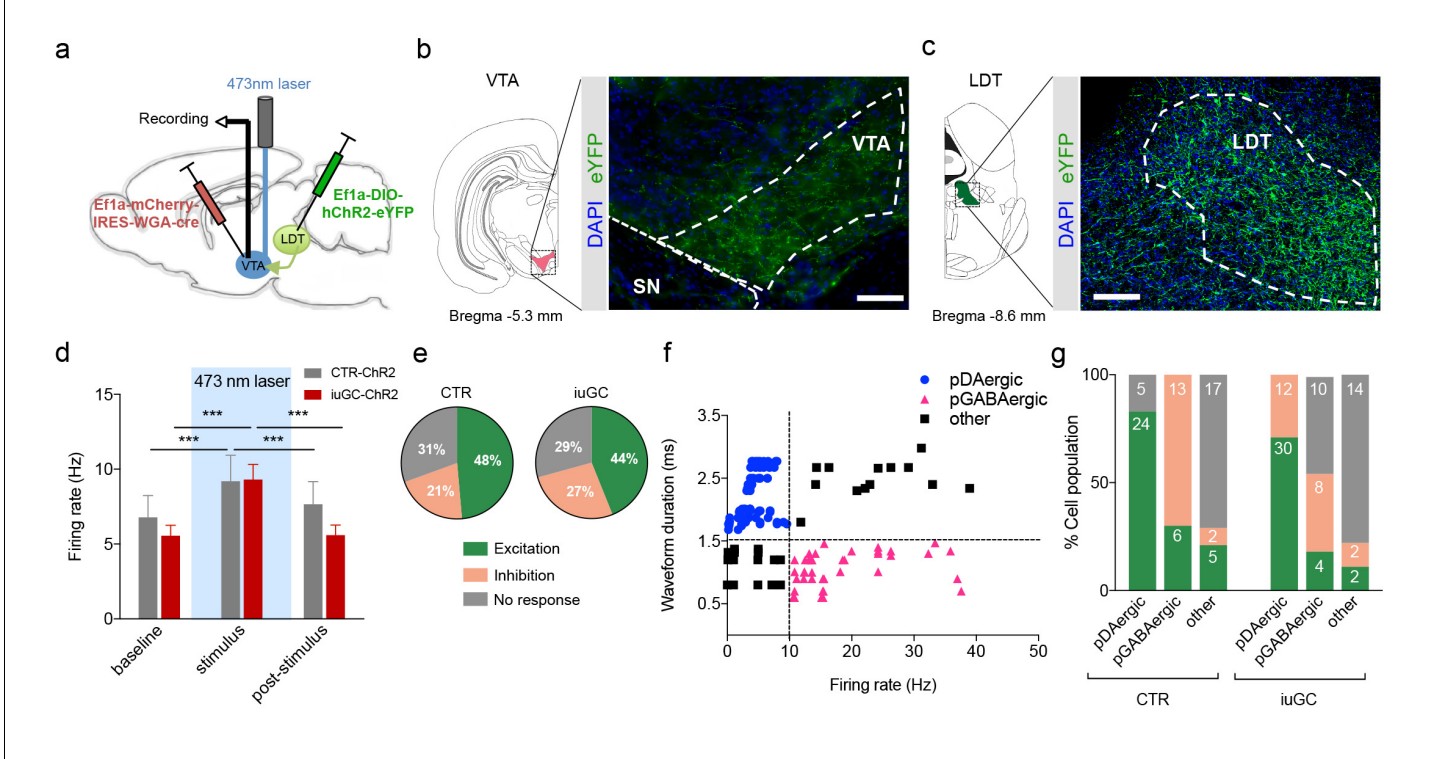

**Figure 3.** Optogenetic activation of LDT terminals in VTA elicits a differential electrophysiological response in iuGC animals. (**a**) Strategy used for optogenetic activation of LDT projecting neurons in the VTA. An AAV5–EF1a–WGA–Cre–mCherry virus construct was injected unilaterally in the VTA, and a cre-dependent ChR2 vector (AAV5-EF1a-DIO-hChR2-eYFP) in the LDT. WGA-Cre will retrogradely migrate and induce the expression of ChR2 in LDT neurons that directly project to the VTA. (**b**) Representative image of immunofluorescence for GFP showing LDT axon terminals in the VTA and (**c**) cell bodies in the LDT; scale bar: 200 μm. (**d**) Optogenetic stimulation of LDT terminals in the VTA (blue rectangle; 30 pulses of 15 ms at 20 Hz) increases the firing rate of VTA neurons in both groups (n_CTR = 72 cells; n_iuGC = 82 cells). (**e**) In CTR, upon LDT terminal stimulation, 48% of recorded VTA cells present an increase in firing rate (of those 69% pDAergic, 17% pGABAergic), 21% decrease activity (0% pDAergic, 87% pGABAergic) and 31% presented no change. In iuGC animals, upon LDT terminal stimulation, 44% of recorded VTA cells present an increase in firing rate (83% pDAergic; 11% GABAergic) 27% decrease activity (55% pGABAergic, 36% DAergic) and 29% presented no change. (**f**) Firing rate and waveform duration were used to classify single units into 3 types of neurons. (**g**) Percentage of each putative neuronal population presenting excitation, inhibition or with no response to LDT terminals optogenetic stimulation. Numbers in bars represent number of cells in each category. pDAergic: putative dopaminergic neurons; pGABAergic: putative GABAergic neurons. Data represented as mean ±s.e.m. ***p<0.001. Additional data is depicted in *Figure 1—figure supplement 1*.

DOI: https://doi.org/10.7554/eLife.25843.007

The following figure supplement is available for figure 3:

**Figure supplement 1.** iuGC treatment has no effect on the response latency after optical stimulation of the LDT-VTA circuit.

DOI: https://doi.org/10.7554/eLife.25843.008

iuGC animals, 44% of cells presented increased firing rate upon stimulation, and the majority were pDAergic neurons (83%). Surprisingly, and clearly different from CTR animals, 55% of cells that presented inhibitory responses were pDAergic neurons and 36% were considered to be pGABAergic neurons (*Figure 3g*). Again, and in accordance with the electrical stimulation data, our optogenetic results suggest an imbalance in the excitatory and inhibitory inputs from the LDT to the VTA.

## Activation of LDT terminals in the VTA rescues motivational deficits of iuGC animals

Since the LDT-VTA circuitry has been described to contribute for positive reinforcement (*Lammel et al., 2012*; *Steidl and Veverka, 2015*; *Lammel et al., 2011*), we evaluated the motivational drive by testing willingness to work for food in a progressive ratio (PR) schedule of reinforcement. This test measures the breakpoint or maximum effort rats are willing to perform for an outcome, when the demand grows progressively over a session.

Training was similar between CTR, CTR-YFP, CTR-ChR2 and iuGC-ChR2 groups across days either in the continuous reinforcement (CRF) or fixed ratio (FR) sessions (*Figure 4—figure supplement 1a–b*). In the test day, iuGC-ChR2 rats presented a significant decrease in breakpoint in comparison to CTR, CTR-YFP and CTR-ChR2 animals (*Figure 4a*; 48,9% decrease; *post-hoc* Bonferroni test CTR vs. iuGC-ChR2: $t_{(68)}$ = 2.882, p=0.0317; CTR-YFP vs. iuGC-ChR2: $t_{(68)}$ = 2.78, p=0.0421; CTR-ChR2 vs. iuGC-ChR2: $t_{(68)}$ = 4.141, p=0.0006), with no differences in the number of pellets earned during the test (*Figure 4—figure supplement 1c*).

We next assessed if selective optogenetic activation of the LDT-VTA pathway was sufficient to enhance motivation during the PR test session. We decided to stimulate animals during cue exposure period since previous work from our group suggested that iuGC animals presented deficits in the Pavlovian-to-Instrumental Transfer test (PIT) (*Soares-Cunha et al., 2014*; *Soares-Cunha et al., 2016*), which measures the ability of a Pavlovian conditioned stimulus that is associated with a reward to invigorate instrumental responding for that (or other) reward (*Corbit and Balleine, 2005*; *Corbit and Janak, 2007*; *Holmes et al., 2010*).

Activation of LDT terminals in the VTA during cue exposure period (30 pulses of 15 ms at 20 Hz; around 15 stimulations per session) reverted the breakpoint of iuGC animals but had no effect in CTR, CTR-eYFP and CTR-ChR2 animals (*Figure 4a–c*; *post-hoc* Bonferroni test CTR: $t_{(34)}$ = 0.1836, p>0.9999; CTR-ChR2: $t_{(34)}$ = 1.203, p=0.9498; CTR-eYFP: $t_{(34)}$ = 0.5099, p>0.9999; iuGC-ChR2: $t_{(34)}$ = 5.007, p<0.0001;). No effect was observed in the number of pellets earned (*Figure 4—figure supplement 1c*). Moreover, no effects in locomotion or free feeding consumption were observed using the same stimulation parameters in either group (*Figure 4—figure supplement 1e–f*).

Importantly, if the LDT-VTA optogenetic activation occurred during the inter-trial interval (ITI) period of the test session, it did not revert iuGC-ChR2 motivational deficits (*Figure 4b–c*; *post-hoc* Bonferroni: $t_{(34)}$ = 0.5138, p>0.9999), suggesting that LDT-VTA activation elicits a positive behavioral response only when it occurs during specific periods of the test.

Because the results of control animals were surprising in the light of previous evidence showing that LDT-VTA stimulation was reinforcing (*Lammel et al., 2012*; *Xiao et al., 2016*), we performed another stimulation protocol in a new set of animals to test this hypothesis (80 pulses of 15 ms at 20 Hz) (*Figure 4—figure supplement 2a–b*). Importantly, when we increased the number of pulses, we observed an increase in the breakpoint of CTR-ChR2 animals (*post-hoc* Bonferroni: $t_{(22)}$ = 3.666, p=0.0054) when compared to breakpoint of CTR or CTR-eYFP animals (*post-hoc* Bonferroni CTR-ChR2 vs. CTR: $t_{(44)}$ = 3.075, p=0.0217; CTR-ChR2 vs. CTR-eYFP: $t_{(44)}$ = 3.194, p=0.0156). Additionally, we observed that this stimulation also increased the breakpoint of iuGC animals (*post-hoc* Bonferroni: $t_{(22)}$ = 4.641, p=0.0005), with no effect in CTR and CTR-eYFP groups (*post-hoc* Bonferroni CTR: $t_{(22)}$ = 0.116, p>0.9999; CTR-eYFP: $t_{(22)}$ = 0.4163, p>0.9999).

## Stimulation of LDT-VTA terminals induces place preference

To get further insight on the role of the LDT-VTA circuit in behavior, we also evaluated the impact of the stimulation of LDT-VTA terminals in the conditioned place preference (CPP) test, which measures the reinforcing capacities of a particular stimulus (*Figure 4d*). LDT-VTA stimulation (30 pulses of 15 ms at 20 Hz every 60 s) elicited conditioning in iuGC group, given by increased preference for the stimulus-associated chamber, ON side, (*Figure 4e–f*; *post-hoc* Bonferroni: $t_{(20)}$ = 5.892, p<0.0001), whereas it did not shift preference in CTR and CTR-eYFP animals.

However, if we increase the number of stimulus (80 pulses of 15 ms every 15 s), we were able to induce conditioning in CTR-ChR2 animals (*Figure 4—figure supplement 2c*; $t_{(8)}$ = 4.737, p=0.0015). Additionally, we observe the same effect in the iuGC-ChR2 group (*Figure 4—figure supplement 2d*; *post-hoc* Bonferroni CTR-ChR2: $t_{(23)}$ = 3.576, p=0.0064; iuGC-ChR2: $t_{(23)}$ = 2.761, p=0.044). No effect was found in CTR and CTR-eYFP animals, as expected (*post-hoc* Bonferroni CTR: $t_{(23)}$ = 0.04378, p>0.9999; CTR-eYFP: $t_{(23)}$ = 0.1521, p>0.9999).

In order to further explore the reinforcing feature of LDT-VTA stimulation, we performed the real-time place preference test (RTPP; *Figure 4g*), where one chamber is paired with optical stimulation and the other is not. Every time the animal was in the designated stimulation box (ON side), it received optical stimulation (15 ms pulses at 20 Hz) that only ended when the animal crossed to the no-stimulation box (OFF side). This test is different from a classic CPP since the animal is able to choose the chamber throughout the test.

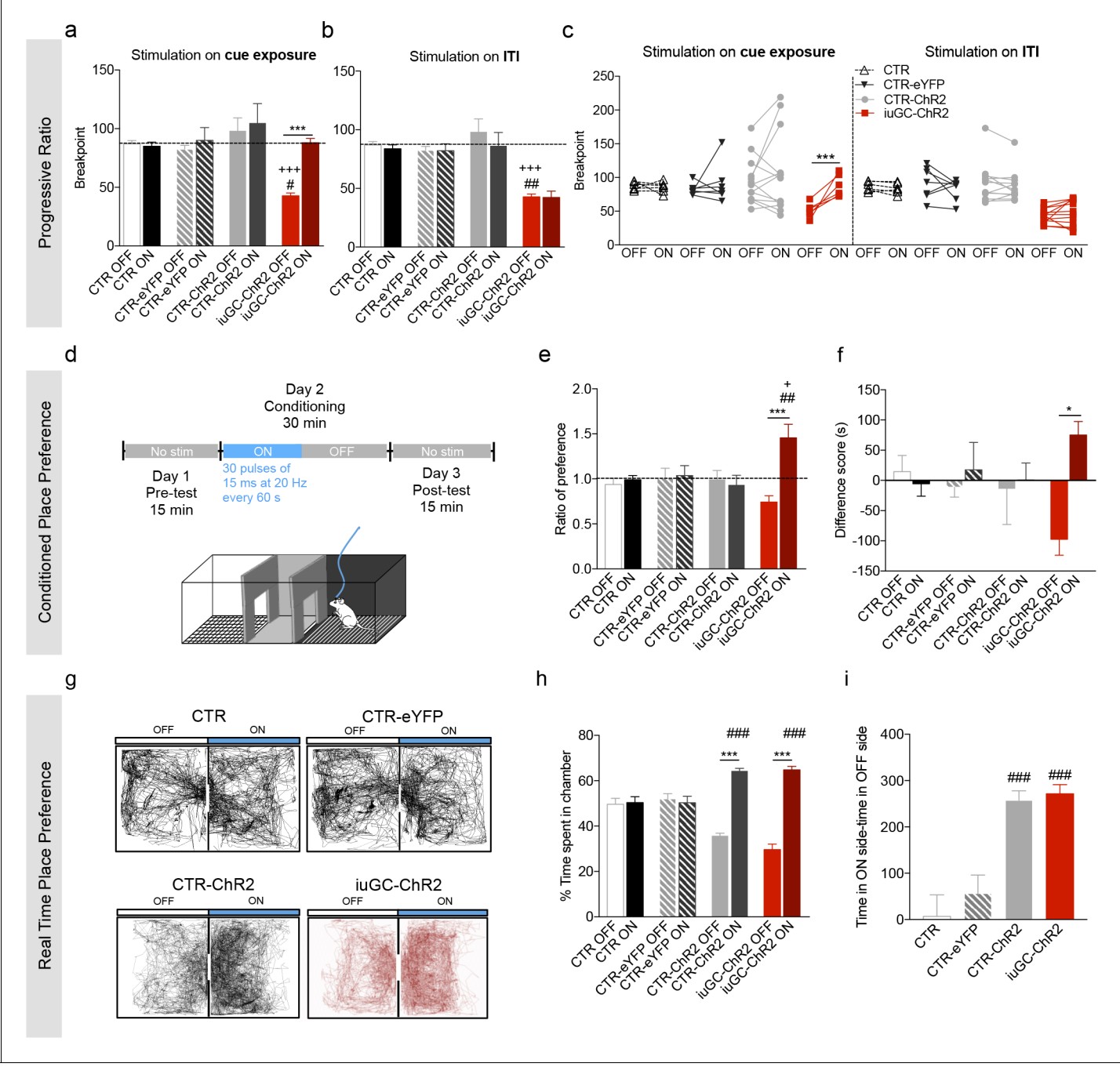

**Figure 4.** Optogenetic activation of LDT-VTA rescues motivational deficits of iuGC-ChR2 animals and induces conditioning. (**a**) Optogenetic stimulation of LDT terminals in the VTA during cue exposure (30 pulses of 15 ms at 20 Hz) rescues the breakpoint deficits in the PR test of iuGC-ChR2 animals, with no effect in other groups ($n_{CTR}$ = 6; $n_{CTR-eYFP}$ = 7; $n_{CTR-ChR2}$ = 13; $n_{iuGC-ChR2}$ = 12). (**b**) Activation of LDT terminals in the VTA in an *irrelevant* period, such as for example during inter-trial interval (ITI) does not change breakpoint of iuGC-ChR2 animals. (**c**) Individual performance in the PR test. All iuGC-ChR2 animals increase their breakpoint when stimulation is associated with the cue but not during the ITI. (**d**) Schematic representation of the CPP protocol. Laser stimulation (30 pulses of 15 ms at 20 Hz, every 60 s) is associated to one chamber. (**e**) Optogenetic stimulation of LDT terminals in the VTA increases preference for the stimulation-paired box (ON) in iuGC-ChR2 but not in CTR-eYFP nor CTR-ChR2 animals ($n_{CTR}$ = 6; $n_{CTR-eYFP}$ = 7; $n_{CTR-ChR2}$ = 5; $n_{iuGC-ChR2}$ = 6). (**f**) Difference score of CPP protocol shown as the difference in time spent in pre- and post-test. iuGC-ChR2 animals present a shift in preference for the ON chamber. (**g**) Real Time Place Preference (RTPP) protocol: animals were placed in a box with two identical chambers for 15 min and allowed to freely explore. When animals crossed to the ON side, optical stimulation was given until exiting the chamber. Shown are representative tracks from a CTR, CTR-eYFP, CTR-ChR2 and an iuGC-ChR2 animal. (**h**) CTR-ChR2 and iuGC-ChR2 rats spend a significantly higher percentage of time in the stimulation-associated box (ON side) ($n_{CTR}$ = 6; $n_{CTR-eYFP}$ = 7; $n_{CTR-ChR2}$ = 8; $n_{iuGC-ChR2}$ = 6). (**i**) Difference between time spent

*Figure 4 continued on next page*

*Figure 4 continued*

in the ON versus OFF side. Data represented as mean ±s.e.m. *p<0.05, **p<0.001, ***p<0.0001. #: comparison with CTR-eYFP; +: comparison with CTR-ChR2. Additional data is depicted in *Figure 4—figure supplements 1–3*.

DOI: https://doi.org/10.7554/eLife.25843.009

The following figure supplements are available for figure 4:

**Figure supplement 1.** Effects of iuGC treatment or optogenetic activation of the LDT-VTA circuit in operant learning, food consumption and locomotion.

DOI: https://doi.org/10.7554/eLife.25843.010

**Figure supplement 2.** Behavioral effects of higher stimulation of LDT-VTA terminals.

DOI: https://doi.org/10.7554/eLife.25843.011

**Figure supplement 3.** Optic fiber placement of animals used for behavioral experiments.

DOI: https://doi.org/10.7554/eLife.25843.012

We observed that stimulation of LDT terminals in the VTA was sufficient to elicit preference for the stimulation-paired chamber in both CTR-ChR2 and iuGC-ChR2 groups (*Figure 4h*; *post-hoc* Bonferroni CTR-ChR2: $t_{(23)}$ = 8.212, p<0.0001; iuGC-ChR2 $t_{(23)}$ = 8.748, p<0.0001) with no effect on control groups (CTR and CTR-eYFP) (*post-hoc* Bonferroni CTR: $t_{(23)}$ = 0.1981, p>0.9999; CTR-eYFP: $t_{(23)}$ = 1.784, p>0.9999). Both CTR-ChR2 and iuGC-ChR2 groups spent significantly more time in the ON side as assessed by the difference of total time spent in each chamber (*Figure 4i*; *post-hoc* Bonferroni CTR-ChR2 vs. CTR: $t_{(23)}$ = 5.451, p=0.0001; CTR-ChR2 vs. CTR-eYFP: $t_{(23)}$ = 4.562, p=0.0008; iuGC-ChR2 vs. CTR: $t_{(23)}$ = 5.387, p<0.0001; iuGC-ChR2 vs. CTR-eYFP: $t_{(23)}$ = 4.543, p=0.0009).

## Discussion

Here we show that prenatal exposure to GCs alters the number of ChAT[+] cells and induces long-lasting expression changes on cholinergic markers (ChAT and AChE) in the LDT, but had no effect on glutamatergic or GABAergic markers. These findings are particularly interesting because both ChAT and AChE contain a glucocorticoid response element (GRE) in their gene *loci*, pinpointing a direct transcriptional regulation by GCs (*Finkelstein et al., 1985*; *Gilad et al., 1985*; *Imperato et al., 1989*; *Berse and Blusztajn, 1997*; *Battaglia and Ogliari, 2005*), although this remains to be confirmed. In fact, it has been shown that stress changes cholinergic enzymes expression (*Finkelstein et al., 1985*; *Kaufer et al., 1998*), either by inducing an alternative splicing of AChE gene (*Nijholt et al., 2004*), or by modulating the epigenetic status of its promoter regions (*Sailaja et al., 2012*). These cholinergic changes observed in iuGC group are more likely to derive from a new equilibrium set *in utero* by GC exposure rather than by changes in the hypothalamic–pituitary–adrenal (HPA) axis, since in adulthood iuGC animals present normal basal levels of corticosterone (*Blaha and Winn, 1993*).

Importantly, this GC *programming effect* in gene expression has been previously demonstrated for other circuits. For example, GC-exposed animals displayed differential methylation status of dopamine receptor D2 promoter region, accompanied by long-lasting gene/protein expression changes in the NAc (*Rodrigues et al., 2012*). These results show that a brief exposure to GC during critical developmental periods may induce persistent effects in specific genes, which may contribute for the increased vulnerability for emotional disorders observed in early life stress models (*Borges et al., 2013a*; *Borges et al., 2013b*; *Piazza and Le Moal, 1996*; *Murgatroyd et al., 2009*).

We also found that the LDT presents decreased basal neuronal activity, and that LDT electrical stimulation produces a differential response in the VTA of iuGC animals. Indeed, iuGC group presented a decrease in the magnitude and duration of excitatory responses in the VTA, and inversely, the inhibitory response was increased, suggesting that the function of the LDT-VTA pathway was compromised. To our knowledge, this is the first report showing electrophysiological differences in the LDT-VTA circuitry induced by GCs. The latency of excitatory responses in the VTA upon LDT electrical (or optogenetic) stimulation in control animals was remarkably low, but we were unable to find any electrophysiological studies to compare to.

Importantly, the latency of VTA excitatory responses to LDT electrical stimulation was substantially increased in iuGC animals. A combination of pre- and post-synaptic iuGC-induced changes may contribute for this phenomenon, however, because this delay is not observed upon optical

excitation of LDT-VTA terminals, it pinpoints to changes in axonal conductivity. Additional studies are now needed in order to understand how GC induces these long-lasting electrophysiological changes.

The VTA contains around 65% of DAergic neurons and 35% of non-DAergic neurons (*Nair-Roberts et al., 2008*), being the latter mainly GABAergic, though subpopulations of glutamatergic neurons as well as dopamine/glutamate co-releasing neurons have been identified (*Yamaguchi et al., 2011*; *Hnasko et al., 2012*). Considering this, we sub-divided the VTA recorded cells into putative DAergic and GABAergic neurons based on their waveform pattern (*Ungless et al., 2004*; *Ungless and Grace, 2012*; *Totah et al., 2013*), yet, it is important to refer that there is still some controversy regarding this categorization (*Margolis et al., 2006*). Importantly, Omelchenko and colleagues suggested that the LDT mediates a divergent excitation/inhibition influence on mesoaccumbens neurons that is likely to excite DAergic cells and inhibit GABA neurons of this region (*Omelchenko and Sesack, 2005*; *Omelchenko and Sesack, 2006*); which is in accordance with our data in control animals.

The LDT provides the tonic input necessary for maintaining burst firing of DAergic neurons (*Lodge and Grace, 2006*) and dopamine release to the NAc (*Blaha et al., 1996*; *Forster and Blaha, 2000*), contributing to reward behaviors (*Lammel et al., 2012*; *Steidl and Veverka, 2015*; *Xiao et al., 2016*). In fact, phasic activation of VTA DAergic neurons can induce behavioral conditioning (*Tsai et al., 2009*) and facilitate positive reinforcement (*Adamantidis et al., 2011*; *Witten et al., 2011*). Conversely, VTA GABAergic neurons provide local inhibition of DAergic neurons (but also long-range inhibition of projection regions, including the NAc), and their activation disrupts reward consummatory behavior (*van Zessen et al., 2012*). Surprisingly, in iuGC animals we observe a shift in the LDT-VTA evoked responses: an increase of inhibition of DAergic neurons and simultaneous decrease of inhibition of GABAergic neurons. This suggests that VTA DAergic neurons are less active, which is in accordance with the observed decreased basal levels of dopamine in the NAc of iuGC animals (*Leão et al., 2007*; *Rodrigues et al., 2012*).

Confirming the functional relevance of the abovementioned electrophysiological data, we have previously shown that iuGC animals exhibited impaired cue-driven motivational drive (*Soares-Cunha et al., 2014*; *Soares-Cunha et al., 2016*). In line with this, we found that iuGC animals presented significant motivational deficits in the PR test. Remarkably, brief optogenetic activation of LDT-VTA terminals during cue exposure was sufficient to rescue the motivation of iuGC animals, with no major impact on control animals, proving that iuGC exposure induces changes in this circuit that are translated into motivational deficits. However, when LDT-VTA stimulation was done during the time-out period of the test, it did not induce any behavioral effect, reinforcing the importance of specific time windows for the stimulation. To our knowledge, this is the first report showing a role of the LDT-VTA circuit in the control of cue-induced motivation. It is important to refer that we decided to use a strategy that activates all LDT inputs because although the majority of its neurons are cholinergic, there are also glutamatergic and GABAergic neurons in the LDT (*Xiao et al., 2016*; *Wang and Morales, 2009*), and each provide parallel sources of input to the VTA. Certainly, additional studies are needed to dissect and evaluate the contribution of each LDT neuronal population for this type of behaviors.

To further understand the role of LDT-VTA in reward behaviors, we tested the animals in two different conditioning paradigms, the non-contingent CPP and the contingent RTPP. Activation of LDT-VTA specific projections shifts animal's preference for the stimulus-associated chamber in both tests. However, iuGC animals seem more susceptible to these reinforcing effects because a *lower stimulation* protocol (30 pulses of 15 ms at 20 Hz) was able to shift iuGC group preference but had no effect in control animals. Importantly, this vulnerability to rewarding/reinforcing stimulus is in accordance with previous data from our team showing that iuGC animals presented increased morphine-associated CPP in comparison to controls (*Rodrigues et al., 2012*). It is thus tempting to speculate that the increased vulnerability of iuGC animals to the effects of LDT-VTA stimulation is due to an imbalance in the excitation-inhibition responses in the VTA triggered by LDT inputs.

In summary, iuGC exposure leads to long-lasting molecular and physiological alterations in the LDT-VTA circuit in parallel with prominent motivational deficits, which were rescued by optogenetic activation of the LDT-VTA terminals. Moreover, we showed that activation of LDT-VTA inputs is reinforcing and that iuGC animals appear to be more vulnerable to the reinforcing properties of this

stimulus. Further studies are now needed to identify how GCs lead to functional changes in vulnerable regions such as the LDT and how this translates into altered behavior.

## Materials and methods

### Animals and treatments

Pregnant Wistar rats were individually housed under standard laboratory conditions (light/dark cycle of 12/12 hr; 22°C); food and water *ad libitum*. Subcutaneous injections of a synthetic GC, dexamethasone (DEX, Sigma, Germany) at 1 mg kg$^{-1}$ (iuGC animals) or vehicle (sesame oil, Sigma, Germany; CTR- control animals) were administered on gestation days 18 and 19 (details of the model can be found in *Leão et al., 2007*; *Borges et al., 2013a*; *Soares-Cunha et al., 2014*; *Rodrigues et al., 2012*; *Borges et al., 2013b*; *Blaha and Winn, 1993*). This model, named iuGC (from *in utero* exposure to GCs) partially mimics the clinical administration of GCs on women in risk of preterm labour (~8% of pregnancies) to promote fetal lung maturation or to manage congenital adrenal hyperplasia during pregnancy. On postnatal day 21, progeny was weaned according to prenatal treatment and gender. Male offspring derived from at least 4 different litters were used.

All manipulations were conducted in strict accordance with European Regulations (European Union Directive 2010/63/EU). Animal facilities and the people directly involved in animal experiments were certified by the Portuguese regulatory entity – DGAV. All the experiments were approved by the Ethics Committee of the University of Minho (SECVS protocol #107/2015). The experiments were also authorized by the national competent entity DGAV (#19074).

### Macrodissection and molecular analysis

Rats were anaesthetized with sodium pentobarbitone (Eutasil, Sanofi, CEVA, Algés, Portugal), decapitated, and heads were immediately snap-frozen in liquid nitrogen. Brain areas of interest were rapidly dissected on ice under a magnifier following specific anatomical landmarks (*Paxinos and Watson, 2007*).

For real-time PCR analysis, total RNA was isolated from samples using Trizol (Invitrogen, Carlsbad, CA, USA) and treated using DNase (Fermentas, Burlington, Canada) according to the manufacturer's instructions. cDNA was synthetized using the iSCRIPT kit (Biorad, Hercules, CA, USA). PCR was performed using EVAGreen SMX (Biorad, Hercules, CA, USA) and the Biorad q-PCR CFX96 apparatus (Biorad, Hercules, CA, USA). *Hprt* was used as housekeeping gene. Relative quantification was used to determine fold changes (control vs. iuGC), using the ΔΔCT method.

Primer sequences used for transgene expression quantification were:

ChAT: Forward, 5'-TCATTAATTTCCGCCGTCTC-3', Reverse, 5'-CCGGTTGGTGGAGTCTTTTA-3'; AChE: Forward, 5'-CCAGAGACAGAGGACATTCTGA-3', Reverse, 5'-GCGTTCCTGCTTGCTATAG TG-3'; VAChT: Forward, 5'-AGTGCCTACTTGGCCAACAC-3', Reverse, 5'-GTCGTAGCTCATGCGA TCAA-3'; EAAC1; Forward, 5'-CATCCCTCATCCCACATCCG-3', Reverse, 5'-CTACCACGA TGCCCAGTACC-3'; GAD1/Gad67: Forward, 5'- GCTCCCTGTGGCTGAATCG-3', Reverse, 5'- GTCC TTTGCAAGAAACCACAG-3'; GAD2/Gad65: Forward, 5'-CTGGCTTTTGGTCCTTCGGA-3', Reverse, 5'-AGCAGAGCGCATAGCTTGTT-3'; GR: Forward, 5'- AGGCCGGTCAGTGTTTTCT-3', Reverse, 5'-CAATCGTTTCTTCCAGCACA-3'.

For western blotting analysis, samples were prepared as previously described (*Rodrigues et al., 2012*). 30 µg of the protein was run in SDS-polyacrylamide gel and then transferred to nitrocellulose membranes. Membranes were incubated with one of the primary antibodies: goat anti-choline acetyltransferase (ChAT, 1:500, Millipore, MA, USA), goat anti-acetylcholine esterase (AChE, 1:500, Abcam, Cambridge, UK), mouse anti-glutamate transporter, excitatory amino-acid transporter, EAAC1 (EAAC1, 1:200, Millipore, MA, USA), rabbit anti-glutamic acid decarboxylase (GAD) 65 + GAD 67 (GAD65/67, 1:10000, Abcam, Cambridge, UK), rabbit anti-glucocorticoid receptor (GR, 1:500, Santa Cruz, CA, USA) and mouse anti-GAPDH (1:200, Iowa, USA) or mouse anti-beta actin (1:2500, Abcam, Cambridge, UK) were used as loading controls for ChAT or AChE, and EAAC1, GAD65/67 or GR, respectively. The secondary antibodies were incubated at a 1:10000 (anti-mouse), 1:5000 (anti-rabbit) and 1:7500 (anti-goat) dilution (Santa Cruz Biotechnologies, Santa Cruz, CA, USA). Membranes were stripped for 15 min at room temperature in stripping buffer (Restore PLUS Western Blot Stripping Buffer, Thermo Scentific, IL, USA) and re-blocked and re-incubated.

Detection was performed using ECL kit (Biorad, Hercules, CA, USA) and bands were quantified using ImageJ (http://rsbweb.nih.gov/ij/).

## Immunohistochemistry (IHC)

Animals were anaesthetized with sodium pentobarbitone (Eutasil, Lisbon, Portugal) and transcardially perfused with saline followed by 4% paraformaldehyde. Brains were removed and sectioned coronally at a thickness of 50 μm, on a vibrating microtome (VT1000S, Leica, Germany).

Free-floating sections were pre-treated with 3% $H_2O_2$ in PBS for 30 min. After blocking using 2.5% fetal bovine serum (FBS) in PBS-Triton 0.3% for 2 hr at room temperature, sections were incubated overnight at 4°C with primary antibody anti-ChAT (1:1000; Millipore, MA, USA). Afterwards, sections were washed and incubated with the secondary polyclonal swine anti-goat biotinylated antibody (1:200, DAKO, Denmark) for 1 hr, and processed with an avidin-biotin complex solution (ABC-Elite Vectastain reagent; Vector Lab., USA) and detected with 0.5 mg ml$^{-1}$ 3,3′-diaminobenzidine (Sigma, Germany) including 12.5 μl of 30% $H_2O_2$ as a substrate in Tris-HCL solution. Sections were washed and mounted on glass slides, air-dried, counterstained with Hematoxilin and coverslipped with Entellan (Merck, NJ, USA). Cell density estimation was obtained by normalizing ChAT$^+$ cells in the corresponding area, determined using an Olympus BX51 optical microscope and the StereoInvestigator software (Microbrightfield). For each animal, 5 slices containing the LDT were used - coordinates according to Paxinos and Watson (*Blaha and Winn, 1993*). The distance of the LDT region analyzed from bregma ranged from: −8.16 mm to −9.48 mm.

## In vivo electrophysiology recordings and stimulation

Animals were anesthetized and submitted to a stereotaxic surgery for the placement of the stimulating and recording electrodes, following anatomical coordinates (*Paxinos and Watson, 2007*). Surgeries were performed under sodium pentobarbitone anaesthesia (induction: 60 mg kg$^{-1}$; maintenance: 15–20 mg kg$^{-1}$, intraperitoneal, Eutasil, Sanofi, CEVA, Algés, Portugal); body temperature was maintained at approximately 37°C with a homoeothermic heat pad system (DC temperature controller, FHC, ME, USA). Anaesthesia level was assessed by observation of pupil size, general muscle tone and by assessing withdrawal responses to noxious pinching.

Stimulating and recording electrodes were placed in the following coordinates: LDT: −8.5 from bregma, 0.8 lateral from midline, −5.5 to −7.9 ventral to brain surface; VTA: −5.4 from bregma, 0.6 lateral from midline, −7.5 to −8.2 ventral to brain surface. A reference electrode was fixed in the skull, in contact with the dura.

Extracellular neural activity from the LDT and the VTA was recorded using a recording electrode (3–7 MΩ at 1 kHz). Recordings were amplified and filtered by the Neurolog amplifier (NL900D, Digitimer Ltd, UK) (low-pass filter at 500 Hz and high-pass filter at 5 kHz). Bi-polar concentric electrode (0.05–0.1 MΩ, Science Products) was inserted in the LDT region. Spontaneous activity of single neurons was recorded to establish baseline for at least 100 s. The stimulation was administered using a square pulse stimulator and a stimulus isolator (DS3, Digitimer, UK). The stimulation consisted of 100 pulses of 0.5 Hz with 0.5 ms duration with intensity from 0.2 to 1 mA. Spikes of a single neuron were discriminated, and data sampling was performed using a CED micro 1401 interface and SPIKE 2 software (Cambridge Electronic Design, Cambridge, UK). Single pulses were delivered to the specific brain region every 2 s. At least 100 trials were administered per cell.

For data analysis, peristimulus time histograms (PSTHs; 5 ms bin width) of neuronal activity were generated during electrical stimulation of the LDT, for each neuron recorded in the VTA. PSTHs were analysed to determine excitatory and inhibitory epochs. Briefly, the mean and standard deviation (SD) of counts per bin were determined for a baseline period, definite as the 500 ms epoch previous stimulation. The onset of excitation was defined as the first of five bins whose mean value exceeded mean baseline activity by 2 SD, and response offset was determined as the time at which activity had returned to be consistently within 2 SD of baseline. Response magnitudes for excitation were calculated with the following equation: (counts in excitatory epoch) - (mean counts per baseline bin 3 number of bins in excitatory epoch). The onset of inhibition was defined as the first of 5 bins whose mean value were below 30% of the baseline activity and the response offset when the activity of the neurons was consistently above 30% of the baseline activity. The total duration of the inhibition was determined for each neuron. We classified single units in the VTA into three separated

groups of putative neurons: putative dopamine (DA), putative GABA, and 'other' neurons. This classification was based on firing rate and waveform duration (calculated from average spike waveform) (*Ungless et al., 2004*; *Ungless and Grace, 2012*; *Totah et al., 2013*). Cells presenting a firing rate <10.0 Hz and a duration of >1.5 ms were considered putative DAergic (pDAergic) neurons. If the firing rate was >10.0 Hz and waveform duration <1.5 ms, cells were assigned to putative GABAergic (pGABAergic) neuron group. Other single units were assigned to the 'other' neuron group. This group likely contains units from both DA and GABA groups.

Regarding the experiments with optical stimulation, a recording electrode coupled with a fiber optic patch cable (Thorlabs) was placed in the VTA or LDT. The DPSS 473 nm laser system (CNI), controlled by a stimulator (Master-8, AMPI), was used for intracranial light delivery and fiber optic output was pre-calibrated to 10–15 mW. Spontaneous activity was recorded for 60 s to establish baseline activity. Optical stimulation consisted of 30 pulses of 15 ms at 20 Hz and 80 pulses of 15 ms at 20 Hz. Firing rate was calculated for the baseline, stimulation period and post stimulation period (60 s after the end of stimulation). Neurons showing a firing rate increase or decrease by more than 20% from the mean frequency of the baseline period were considered as responsive, as previously reported by Benazzouz and colleagues (*Benazzouz et al., 2000*).

At the end of each electrophysiological experiment, all brains were collected and processed to identify recording region.

### Optogenetics constructs

AAV5–EF1a–WGA–Cre–mCherry, AAV5–EF1a–DIO–hChR2–YFP and AAV5–EF1a–DIO–YFP were obtained directly from the Gene Therapy Center Vector Core (UNC) center (vectors kindly provided by Karl Deisseroth, Stanford University). AAV5 vector titers were 2.1–6.6 $\times$ 10$^{12}$ virus molecules ml$^{-1}$.

### Surgery and cannula implantation

Rats designated for behavioral experiments were anesthetized with 75 mg kg$^{-1}$ ketamine (Imalgene, Merial) plus 0.5 mg kg$^{-1}$ medetomidine (Dorbene, Cymedica). One μl of AAV5–EF1a–WGA–Cre–mCherry was unilaterally injected into the VTA (coordinates from bregma, according to Paxinos and Watson: −5.4 mm anteroposterior, +0.6 mm mediolateral, and −7.8 mm dorsoventral) and 1 μl of AAV5–EF1a–DIO–hChR2–YFP was injected in the LDT (coordinates from bregma: −8.5 mm anteroposterior, +0.9 mm mediolateral, and −6.5 mm dorsoventral) in both CTR and iuGC groups (CTR-ChR2 and iuGC-ChR2). We had two additional groups: a control group (CTR) that was injected only with 1 μl AAV5–EF1a–DIO–hChR2–YFP in the LDT; and CTR-YFP animals which were injected with 1 μl AAV5–EF1a–WGA–Cre–mCherry in the VTA and 1 μl AAV5–EF1a–DIO–YFP in the LDT. Rats were then implanted with an optic fiber (200 μm core fiber optic; Thorlabs, NJ, USA) with 2.5 mm stainless steel ferrule (Thorlabs, NJ, USA) using the injection coordinates for the VTA (with the exception of dorsoventral: −7.7 mm) that were secured to the skull using 2.4 mm screws (Bilaney, Germany) and dental cement (C and B kit, Sun Medical). Rats were removed from the stereotaxic frame and sutured. Anaesthesia was reverted by administration of atipamezole (1 mg/kg). After surgery animals were given anti-inflammatory (Carprofeno, 5 mg/kg) for one day, analgesic (butorphanol, 5 mg/kg) for 3 days, and were let to fully recover before initiation of behavior. Optic fiber placement was confirmed for all animals after behavioral experiments (*Figure 4—figure supplement 3*). Animals that were assigned for electrophysiological experiments were not implanted with an optic fiber.

### Behavior

#### Progressive ratio schedule of reinforcement

Rats were placed and maintained on food restriction ($\approx$7 g/day of standard lab chow) to maintain 90% free-feeding weight. Behavioral sessions were performed in operant chambers (Med Associates, IL, USA) containing a central magazine that provided access to 45 mg food pellets (Bio-Serve), two retractable levers located on each side of the magazine with cue lights above them. A 2.8W, 100mA house light positioned at the top-centre of the wall opposite to the magazine provided illumination. A computer equipped with Med-PC software (Med Associates, IL, USA) controlled the equipment and recorded the data.

The behavioral protocol was previously described (*Soares-Cunha et al., 2016*; *Wanat et al., 2013*). Animals were first trained on continuous reinforcement (CRF) schedule: a single lever press

yields one pellet. Side of the active lever was alternated between sessions. Rats were then trained in a fixed ratio (FR) schedule comprising 50 trials with both levers presented, but the active lever signalled by the illumination of the above cue light. When achieving the correct number of lever presses, a pellet was delivered, levers retracted and the cue light turned off for a 20 s inter-trial interval (ITI). Following up, rats were trained using an FR4 reinforcement schedule for 4 days and a FR8 for one day, for both levers. Rats were then exposed to the following schedule: day 1 – FR4 (left lever); day 2- PR (left lever); day 3- FR4 (left lever); day 4 – FR4 (right lever); day 5 – PR (right lever). Food rewards were earned on an FR4 reinforcement schedule during FR sessions. PR sessions were similar to FR4 sessions except the operant requirement on each trial (T) was the integer (rounded down) of $1.4^{(T-1)}$ lever presses, starting at 1 lever press. PR sessions ended after 15 min without completion of the response requirement in a trial.

Before the PR session began, rats were connected to an opaque optical fiber in the VTA through previously implanted fiber optic cannula. The optical fiber was connected to a 473 nm DPSS laser (CNI Laser), controlled using a pulse generator (Master-8; AMPI). At the beginning of each trial of the PR session – when the cue light was turned on – animals received an optical stimulation, which consisted in 30 pulses of 15 ms at 20 Hz (473 nm; 10 mW of light at the tip of the optic fiber). In a second set of animals, the number of pulses was increased to 80 pulses of 15 ms at 20 Hz during each cue exposure. CTR, CTR-YFP, CTR-ChR2 and iuGC-ChR2 received this optical stimulation.

### Conditioned place preference – CPP

The CPP protocol was adapted from a previously published report (*Lammel et al., 2012*; *Ungless and Grace, 2012*). Briefly, on day 1, individual rats were placed in the centre chamber and allowed to freely explore the entire apparatus for 15 min (pre-test). On day 2, rats were confined to one of the side chambers for 30 min and paired with optical stimulation, ON side; in the second session, rats were confined to the other side chamber for 30 min with no stimulation, OFF side. Conditioning sessions were counterbalanced. On day 3 rats were allowed to freely explore the entire apparatus for 15 min (post-test). Optical stimulation consisted of 30 pulses of 15 ms at 20 Hz, every 60 s. In a second set of animals optical stimulation was increased to 80 pulses of 15 ms at 20 Hz, every 15 s.

### Real-time place preference – RTPP

RTPP test was performed in a custom-made black plastic arena (60 × 60 × 40 cm) comprised by two indistinguishable chambers, for 15 min. One chamber was paired with light stimulation of 15 ms pulses at 20 Hz during the entire period that the animal stayed in the stimulus-paired side. The choice of paired chamber was counterbalanced across rats. Animals were placed in the no-stimulation chamber at the start of the session and light stimulation started at every entry into the paired chamber. Animal activity was recorded using a video camera and time spent in each chamber was manually assessed. Results are presented as total time spent in each chamber.

### Immunofluorescence (IF)

Animals were anaesthetized with sodium pentobarbitone (Eutasil, Lisbon, Portugal) and transcardially perfused with 0.9% saline followed by 4% paraformaldehyde. Brains were removed and sectioned coronally at a thickness of 50 μm, on a vibrating microtome (VT1000S, Leica, Germany). Sections were incubated overnight, with the primary antibody goat anti-GFP (1:500, Abcam, Cambridge, UK), followed by secondary fluorescent antibody (1:1000, Invitrogen, MA, USA). All sections were stained with 4',6-diamidino-2-phenylindole (DAPI; 1 mg ml-1) and mounted using mounting media (Permafluor, Invitrogen, MA, USA).

### Statistical analysis

Statistical analysis was performed in GraphPad Prism 5.0 (GraphPad Software, Inc., La Jolla, CA, USA) and SPSS Statistics v19.0 (IBM corp., USA). Parametric tests were used whenever Shapiro-Wilk normality test SW >0.05. Two-way analysis of variance (ANOVA) was used when appropriate. Bonferroni's post hoc multiple comparison tests were used for group differences determination. Statistical analysis between two groups was made using Student's t-test. Results are presented as mean ±SEM. Statistical significance was accepted for $p < 0.05$.

## Acknowledgements

Authors would like to thank Karl Deisseroth from Stanford University for the optogenetic virus. BC, CS-C, and SB are recipients of Fundação para a Ciência e Tecnologia (FCT) fellowships (SFRH/BD/98675/2013; SFRH/BD/51992/2012; SFRH/BD/89936/2012). AJR is a FCT Investigator (IF/00883/2013). This work was co-financed by the Portuguese North Regional Operational Program (ON.2 – O Novo Norte) under the National Strategic Reference Framework (QREN), through the European Regional Development Fund (FEDER). This work was partially financed by BIAL grant 30/16. Part of the work was financed by Projeto Estratégico – LA 26 – 2013–2014 (PEst-C/SAU/LA0026/2013).

## Additional information

### Funding

| Funder | Grant reference number | Author |
| --- | --- | --- |
| Fundação para a Ciência e a Tecnologia | PhD scholaships -FCT investigator position | Bárbara Coimbra<br>Carina Soares-Cunha<br>Sónia Borges<br>Ana João Rodrigues |
| Fundação para a Ciência e a Tecnologia | Projeto Estratégico - LA 26 - 2013-2014 (PEst-C/SAU/LA0026/2013) | Bárbara Coimbra<br>Carina Soares-Cunha<br>Sónia Borges<br>Nivaldo AP Vasconcelos<br>Nuno Sousa<br>Ana João Rodrigues |
| Fundação para a Ciência e a Tecnologia | SFRH/BD/98675/2013 | Bárbara Coimbra<br>Carina Soares-Cunha<br>Sónia Borges |
| Fundação para a Ciência e a Tecnologia | SFRH/BD/51992/2012 | Bárbara Coimbra<br>Carina Soares-Cunha<br>Sónia Borges |
| Fundação para a Ciência e a Tecnologia | SFRH/BD/89936/2012 | Bárbara Coimbra<br>Carina Soares-Cunha<br>Sónia Borges |
| The National Strategic Reference Framework's (QREN) | | Nuno Sousa<br>Ana João Rodrigues |
| O Novo Norte | | Nuno Sousa<br>Ana João Rodrigues |
| BIAL Foundation | 30/2016 | Ana João Rodrigues |

The funders had no role in study design, data collection and interpretation, or the decision to submit the work for publication.

### Author contributions

Bárbara Coimbra, Data curation, Formal analysis, Validation, Investigation, Visualization, Methodology, Writing—original draft, Writing—review and editing; Carina Soares-Cunha, Data curation, Formal analysis, Investigation, Methodology; Sónia Borges, Formal analysis, Visualization; Nivaldo AP Vasconcelos, Data curation, Formal analysis, Investigation; Nuno Sousa, Conceptualization, Supervision, Funding acquisition, Project administration, Writing—review and editing; Ana João Rodrigues, Conceptualization, Supervision, Funding acquisition, Investigation, Methodology, Writing—original draft, Project administration, Writing—review and editing

### Author ORCIDs

Bárbara Coimbra (iD) http://orcid.org/0000-0003-1737-2268
Ana João Rodrigues (iD) https://orcid.org/0000-0003-1968-7968

## Ethics

Animal experimentation: All manipulations were conducted in strict accordance with European Regulations (European Union Directive 2010/63/EU). Animal facilities and the people directly involved in animal experiments were certified by the Portuguese regulatory entity - DGAV. All of the experiments were approved by the Ethics Committee of the University of Minho (SECVS protocol #107/2015). The experiments were also authorized by the national competent entity DGAV (#19074).

## Decision letter and Author response

Decision letter https://doi.org/10.7554/eLife.25843.013
Author response https://doi.org/10.7554/eLife.25843.014

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
