## [Decision Letter]

Thank you for submitting your article "Impairments in laterodorsal tegmentum to VTA projections underlie glucocorticoid-triggered reward deficits" for consideration by *eLife*. Your article has been reviewed by three peer reviewers, and the evaluation has been overseen by a Reviewing Editor and a Senior Editor. The following individual involved in review of your submission has agreed to reveal his identity: Eric Dumont (Reviewer #3).

The reviewers have discussed the reviews with one another and the Reviewing Editor has drafted this decision to help you prepare a revised submission. The reviewers find the work interesting and relevant but raise a few important points that should be addressed. Attached are the point-by-point comments of the reviewers, but below are the more relevant issues to be addressed.

– In the optical stimulation experiments, there should be controls for Cre expression and viral expression. The use of a Cre dependent virus expressing GFP would be advisable.

– The article would benefit from a more detailed description and a more specific use of statistics. For example, a two-way ANOVA with age and glucocorticoid treatment as factors, followed by post-hoc tests seem more appropriate than individual t-test or Mann-Whitney tests for every age. Also, the rationale for the use or not of specific statistical tests, multiple comparison corrections, etc. should be provided.

– Are there differences in GR expression levels in the LDT or corticosterone levels in these animals at different developmental stages that accompany the observed effects on the expression of cholinergic enzymes? Is the observed effect the result of a new equilibrium set after the in-utero treatment, or is it perpetuated through differences in GR expression or corticosterone levels?

– The authors should specify how the loading controls were done for the Westerns in Figure 1, or provide appropriate loading controls.

– The authors should provide more details about the physiology with representative examples, showing recording and stimulation sites, and a few more suggested analyses.

In general, more details are needed in the Materials and methods, such as the volume of injected virus, representative example of microinjections the rationale for using a 20% cut-off for excitation or inhibition should be provided, etc.

Reviewer #1:

This study investigates the role of prenatal glucocorticoids (GCs) on the LDT-VTA circuitry and its impact on reward-related behavior. In their study, the authors use a rat model of in utero GC (iuGC) exposure at gestation days 18 and 19, which shows long-lasting changes in the cholinergic system (i.e., increased number of ChAT^+^ cell density and altered transcript and protein levels of the markers ChAT and AChE) in the LDT. Notably, the study stems from previous findings from the same group published in 2013 (Borges et al., 2013) where, similarly, they reported differences in ChAT^+^ cell density in LDT induced by the same glucocorticoid treatment. Here, AChE expression analysis was added, as well as analyses performed at different time points and expression at the mRNA and protein levels. The characterization of the LDT-VTA circuit with in vivo electrophysiology revealed a decreased basal neuronal activity of the LDT, but not VTA, in iuGC animals. LDT stimulation showed a decreased magnitude of excitation and an increase in VTA inhibition, likely resulting from a shift of the cell-types that respond to the stimulus. Accompanying these described LDT-VTA dysfunctions, the authors demonstrate that iuGC rats exhibit motivational deficits in the willingness to work for food, which were rescued by optogenetic activation of the LDT-VTA pathway. Furthermore, iuGC animals are more susceptible to the reinforcing properties of LDT-VTA stimulation (rats exposed to glucocorticoids in utero were more sensitive to optogenetic stimulation of LDT-VTA projections to develop conditioned place preference). The authors conclude that iuGC exposure affects the LDT-VTA circuit and results in motivational deficits.

The work is interesting and the findings relevant. Nevertheless, there are issues and concerns that require clarification:

1) It is not clear why the work does not follow up on the identified changes on ChAT and AChE directly, as no specific manipulation assessed the role of cholinergic projections from the LDT to the VTA.

2) As raised by the authors themselves, they have carried out their rescue experiments by activating all LDT inputs and not distinguishing between cholinergic, glutamatergic and GABAergic neurons in the LDT. The authors mention in the introduction that there are contributions of glutamatergic projections from the LDT to the VTA; it would make sense to study glutamatergic markers as well to assess whether the effect of prenatal glucocorticoid exposure is specific to the cholinergic input from LDT to VTA or not and bridge the different parts of the work.

3) A two-way ANOVA with age and glucocorticoid treatment as factors, followed by post-hoc tests seem more appropriate than individual t-test or Mann-Whitney tests for every age.

4) Key questions not answered by the study: Are there differences in GR expression levels in the LDT or corticosterone levels in these animals at different developmental stages that accompany the observed effects on the expression of cholinergic enzymes? Is the observed effect the result of a new equilibrium set after the in utero treatment, or is it perpetuated through differences in GR expression or corticosterone levels?

5) In the optical stimulation experiments, control rats were treated only with an AAV-EF1a-DIO-hChR2-YFP virus. How did the authors check for proper viral injection? In addition, there seems to be no control for any effects Cre expression alone might have. The best control would be treatment with the same Cre-expressing virus in the VTA expressing only GFP in the LDT.

6) In Figure 1 representative immunoblot of ChAT and AChE in the LDT is represented followed by quantification of the bands. Why only one GAPDH loading control? Did the authors run an immunoblot for both proteins and then strip their membrane, which would explain one GAPDH control, or did they use two independent membranes for ChAT and AChE? In the latter case, they should please provide also a GAPDH loading control for their second membrane, which they used for the quantification plot in Figure 1. Please provide this information in material and methods.

7) The article would benefit by providing (molecular) background regarding the rationale to analyze cholinergic markers in iuGC animals.

8) The article would benefit from a more detailed description about the statistical tests used for the experiments. Specifically, did they use multiple correction for Figure 1? And for Figure 2? The authors report the p-values in an exemplary manner as APA-Style. Could the authors please also provide these data for Figure 3? In subsection “iuGC treatment impairs the LDT-VTA circuitry”: Please check whether the p-value of 0.008 for the magnitude is correct. It seems improbable given that t(30) = 3.723.

Reviewer #2:

This manuscript by Coimbra and colleagues follows up on several papers by the same group pointing out the effects of iuGC exposure, this time describing motivational deficits in adult rats with the LDT->VTA interaction involved in such deficits. They presented evidence about an increased number of Chat neurons (and an increase in Chat RNA or protein levels) in the LDT from adult rats. They performed recordings in the LDT and VTA to measure the basal level of activity, electrically stimulated in the LDT to record its effect in the VTA, optogenetic stimulated the LDT->VTA terminals in the VTA evaluating the excitation/inhibition responses of VTA. And also show evidence of an altered response in the break point task in the iuGC model, which was reverted by LDT stimulation. Importantly, this last experiment showed temporal specificity of the stimulation.

Based on the evaluation of the evidence presented, this reviewer considers that this manuscript is suitable for publication attending the following points.

Figure 2 and Figure 3

Specify what were the animals doing when the baseline described was acquired.

Present a distribution of the waveform duration vs. the firing rate.

Discuss the mechanisms behind the difference in latencies when comparing the LDT-VTA responses in the control vs. the iuGC groups.

About the statement citing Figure 2: Altogether, these data demonstrated an imbalance in the excitatory and inhibitory inputs from the LDT. I suggest: Altogether, these data demonstrated an imbalance in the excitatory and inhibitory inputs to the VTA when electrically stimulating in the LDT.

Why did the optogenetic stimulation was not performed in the LDT?

Please show images of the LDT cells expressing ChR2.

Why, if Figure 3 shows overall more decreased responses, is this not reflected in Figure 3? May it be by a different probability to record from pDA or pGABAergic units in each group?

Figure 4.

When presenting that activation of LDT terminals in the VTA during cue exposure period normalized the breakpoint of iuGC animals but had no effect in CTR animals, I suggest to place it next to the other protocol (both in text and figures). In this sense, it is clearer to the reader that LDT(ChR2)->VTA stimulation revert the effect on the break point rather than normalizing it.

Figure 4—figure supplement 1.

For the analysis of locomotion in the LDT->VTA please plot the distance aligned to each time of the stimulation: that may reveal a different conclusion (see DOI: 10.1371/journal.pone.0033612).

Reviewer #3 (General assessment and major comments (Required)):

This is an exceptional study combining several state-of-the-art neurobiology approaches and providing a convincing demonstration of the effect of pre-natal stress-like conditions in the neural circuits of motivation. The manuscript is very-well presented, the data are clear, and most controls (positive and negative) have been provided. This was clearly a well-designed series of experiments which ended-up with extremely convincing conclusions. All claims made by the authors was well-supported by the presented data.

Overall, I have no concerns with this study and i believe that it should be published as is.

[Editors' note: further revisions were requested prior to acceptance, as described below.]

Thank you for submitting your article "Impairments in laterodorsal tegmentum to VTA projections underlie glucocorticoid-triggered reward deficits" for consideration by *eLife*. Your article has been reviewed by two peer reviewers, and the evaluation has been overseen by a Reviewing Editor and a Senior Editor. The following individual involved in review of your submission has agreed to reveal his identity: Eric Dumont (Reviewer #3).

The reviewers have discussed the reviews with one another and the Reviewing Editor has drafted this decision to help you prepare a revised submission. Although the reviewers feel that the manuscript has been improved, they think some points need additional clarification before the study can be acceptable for publication.

Reviewer #2:

Based on the evidence presented this reviewer considers that this manuscript is suitable for publication only suggesting further attention to the following points:

Original point: Present a distribution of the waveform duration vs. the firing rate.

Response from the authors: In accordance with the reviewer's suggestion, we have now included additional graphs in Figure 2 and Figure 3 presenting the waveform duration vs. the firing rate of recorded cells.

Original point: Discuss the mechanisms behind the difference in latencies when comparing the LDT-VTA responses in the control vs. the iuGC groups.

Response from the authors: Thank you for raising this point […] Does the reviewer have any idea/suggestion?

Further request to address this point: Previously when requesting to discuss the mechanisms behind the difference in latencies when comparing the LDT-VTA responses in the control versus the iuGC groups I was looking for an explanation based on mechanism as now is provided (perhaps also discussing axonal conductivity, note that when stimulating the LDT axons in the VTA no changes in latency are observed; Figure 3—figure supplement 1), but also a discussion with the pre-existing measurements of this latency (if there is any by other groups), since a 1ms latency seems to fast, calling attention as if the stimulating electrode may not have been positioned exactly in the LDT.

Original point: Why, if Figure 3 shows overall more decreased responses, is this not reflected in Figure 3? May it be by a different probability to record from pDA or pGABAergic units in each group?

Response from the authors: We agree with the reviewer. In fact, the probability of recording pDAergic and pGABAergic units is different in the VTA region. We have a higher recruitment of pDAergic cells upon optical stimulation, which is in accordance with anatomical studies showing the preferential inputs of the LDT to mesoaccumbal DAergic cells in the VTA (Omelchenko and Sesack, 2005).

Original point: Figure 4. When presenting that activation of LDT terminals in the VTA during cue exposure period normalized the breakpoint of iuGC animals but had no effect in CTR animals, I suggest to place it next to the other protocol (both in text and figures). In this sense, it is clearer to the reader that LDT(ChR2)->VTA stimulation revert the effect on the break point rather than normalizing it.

Response from the authors: We are not certain that we understood this comment. Figure 4—figure supplement 1. We have made some changes in the behavioral graphs of the manuscript, please check.

Further request to address this point: The point was that the optogenetic stimulation of the LDT terminals is not specific to normalize the breakpoint of iuGC animals, since activating this terminal using a stronger protocol also increase the break point in either control or iuGC animals.

Original point: For the analysis of locomotion in the LDT->VTA please plot the distance aligned to each time of the stimulation, that may reveal a different conclusion (see DOI: 10.1371/journal.pone.0033612).

Response from the authors: We analyzed the paper mentioned, but we did not fully understand what the reviewer asks for. We have now included an additional graph with total distance travelled. Please check if this is what you asked for.

Further request to address this point: When stating: Moreover, no effects in locomotion […] were observed using the same stimulation parameters in either group (Figure 4—figure supplement 1). This conclusion arises from the analysis performed (one point measured every tens of milliseconds) however an immediate effect on locomotion cannot be discharged (in the range of milliseconds). I previously pointed to the DOI: 10.1371/journal.pone.0033612 because it was one of the few papers at that time recognizing that there are motor modulations when manipulating VTA Dopaminergic cells activity. Please perform a smaller bin analysis of the data evaluating the manipulations on locomotion.

Reviewer #3:

This revised version of the manuscript has been substantially improved and addressed several if not all of my concerns. The manuscript now includes important missing control data (e.g. CRE and viral expression). When appropriate the authors substantiated their rationale and claims.

---

## [Author Response]

*The reviewers have discussed the reviews with one another and the Reviewing Editor has drafted this decision to help you prepare a revised submission. The reviewers find the work interesting and relevant but raise a few important points that should be addressed. Attached are the point-by-point comments of the reviewers, but below are the more relevant issues to be addressed.*

*– In the optical stimulation experiments, there should be controls for Cre expression and viral expression. The use of a Cre dependent virus expressing GFP would be advisable.*

We have performed an additional experiment, including a new control group which consists in the injection of AAV5–EF1a–WGA–Cre–mCherry in the VTA and AAV5–EF1a–DIO–YFP in the LDT – please check answers to reviewers below.

*– The article would benefit from a more detailed description and a more specific use of statistics. For example, a two-way ANOVA with age and glucocorticoid treatment as factors, followed by post-hoc tests seem more appropriate than individual t-test or Mann-Whitney tests for every age. Also, the rationale for the use or not of specific statistical tests, multiple comparison corrections, etc. should be provided.*

We have taken this comment into consideration.

*– Are there differences in GR expression levels in the LDT or corticosterone levels in these animals at different developmental stages that accompany the observed effects on the expression of cholinergic enzymes? Is the observed effect the result of a new equilibrium set after the in-utero treatment, or is it perpetuated through differences in GR expression or corticosterone levels?*

Most likely due to a new equilibrium set after the in-utero treatment, as we discuss below in response to reviewer 1.

*– The authors should specify how the loading controls were done for the Westerns in Figure 1, or provide appropriate loading controls.*

Done – please check answers to reviewers below.

*– The authors should provide more details about the physiology with representative examples, showing recording and stimulation sites, and a few more suggested analyses.*

We believe we have addressed all of the reviewer comments.

*In general, more details are needed in the Materials and methods, such as the volume of injected virus, representative example of microinjections the rationale for using a 20% cut-off for excitation or inhibition should be provided, etc.*

We have added the volume of injected virus (Discussion section); and the reference paper supporting the 20% cut-off decision for electrophysiological recordings (Discussion section). Figure with the optic fiber placement for each animal used in behavioral experiments is now provided (Figure 4—figure supplement 3).

*Reviewer #1:*

*[…] 1) It is not clear why the work does not follow up on the identified changes on ChAT and AChE directly, as no specific manipulation assessed the role of cholinergic projections from the LDT to the VTA.*

Thank you for raising this point, we have now explained better in the manuscript our rationale for activating all inputs and not only cholinergic. We decided to activate all inputs because we observed an effect of iuGC exposure in both excitatory and inhibitory VTA responses elicited by LDT electrical activation, suggesting that different neurotransmitters systems could be involved.

We have now added this information in subsection “Optogenetic activation of LDT terminals in the VTA elicits distinct responses in control and iuGC animals **“**of the revised manuscript.

*2) As raised by the authors themselves, they have carried out their rescue experiments by activating all LDT inputs and not distinguishing between cholinergic, glutamatergic and GABAergic neurons in the LDT. The authors mention in the introduction that there are contributions of glutamatergic projections from the LDT to the VTA; it would make sense to study glutamatergic markers as well to assess whether the effect of prenatal glucocorticoid exposure is specific to the cholinergic input from LDT to VTA or not and bridge the different parts of the work.*

We agree with the reviewer that it is interesting to study the impact of iuGC in other neuronal populations of the LDT. Thus, we have now included gene and protein expression data of glutamatergic and GABAergic markers in the LDT in the revised manuscript (subsection “Sustained cholinergic dysfunction in iuGC animals “; Figure 1—figure supplement 3). We found no major effect in the expression levels of EAAC1, GAD1, GAD2, GAD65 and GAD67.

*3) A two-way ANOVA with age and glucocorticoid treatment as factors, followed by post-hoc tests seem more appropriate than individual t-test or Mann-Whitney tests for every age.*

Considering this comment, we believe that the way we presented these results was not clear enough, thus we decided to separate data according to ages. We presented individual t-test or Mann-Whitney tests for every age, since gene or protein expression (RT-PCR plates or Western Blot membranes) were performed on distinct animals (not repeated measures). Please check new graphs.

*4) Key questions not answered by the study: Are there differences in GR expression levels in the LDT or corticosterone levels in these animals at different developmental stages that accompany the observed effects on the expression of cholinergic enzymes? Is the observed effect the result of a new equilibrium set after the in utero treatment, or is it perpetuated through differences in GR expression or corticosterone levels?*

We agree that this is an interesting point. Previous data from our group has shown no major differences in corticosterone levels in these animals (Oliveira et al., 2006); only when we challenged these animals we observe differences in corticosterone secretion – such as for example in dexamethasone suppression test (Oliveira et al., 2006).

Regarding GR expression, we have performed RT-PCR and western blotting against GR, which presented no significant differences between controls and iuGC animals (Results section; Figure 1—figure supplement 3–L, T–V).

As the reviewer pinpoints, we believe that early life exposure to GCs reprograms the circuit, setting a new equilibrium in the inputs from the LDT to the VTA. We have now included a sentence about this in the Discussion section.

*5) In the optical stimulation experiments, control rats were treated only with an AAV-EF1a-DIO-hChR2-YFP virus. How did the authors check for proper viral injection? In addition, there seems to be no control for any effects Cre expression alone might have. The best control would be treatment with the same Cre-expressing virus in the VTA expressing only GFP in the LDT.*

The CTR group that we used (only injected with AAV5–EF1a–DIO-hChR2–YFP in the LDT) was included in order to confirm the absence of expression of the channelrhodopsin in our approach in wild-type animals, ensuring the need for transsynaptic migration of Cre protein from the VTA in order for ChR2 to be expressed.

Yet, the reviewer raised a pertinent point, therefore, we performed additional experiments to include an additional control group (designated CTR-eYFP) injected with AAV5–EF1a–WGA–Cre–mCherry in the VTA and AAV5–EF1a–DIO–YFP in the LDT. Please check the new version of the manuscript; behavioral data of Figure 4.

*6) In Figure 1 representative immunoblot of ChAT and AChE in the LDT is represented followed by quantification of the bands. Why only one GAPDH loading control? Did the authors run an immunoblot for both proteins and then strip their membrane, which would explain one GAPDH control, or did they use two independent membranes for ChAT and AChE? In the latter case, they should please provide also a GAPDH loading control for their second membrane, which they used for the quantification plot in Figure 1. Please provide this information in material and methods.*

This was our mistake, thank you for pointing this out. We used two different membranes, thus we used two loading controls for quantification. After ChAT and AChE blots, we then stripped each membrane to probe against the loading protein antibody.

Figure is now corrected. We have also included, in the Materials and methods section, the membrane stripping protocol performed in the immunoblots.

*7) The article would benefit by providing (molecular) background regarding the rationale to analyze cholinergic markers in iuGC animals.*

Considering the reviewer comment, we believe that the description of the rationale behind the analysis of cholinergic changes in iuGC animals was, perhaps, not clear enough. Previous results by our group have shown an increase of LDT cholinergic cells recruited after an adverse stimulus (Borges et al., 2013). This led us to evaluate when these cholinergic changes occurred and the functional relevance of such alterations.

We have now included the following text (subsection “Sustained cholinergic dysfunction in iuGC animals“): “Previous data from our team suggested that LDT cholinergic cells were differentially recruited between in response to an adverse stimulus in a model of in utero GC (iuGC) exposure at gestation days 18 and 19. Considering this, we first evaluated the impact of GCs on the cholinergic circuitry of iuGC animals.”.

*8) The article would benefit from a more detailed description about the statistical tests used for the experiments. Specifically, did they use multiple correction for Figure 1? and for Figure 2? The authors report the p-values in an exemplary manner as APA-Style. Could the authors please also provide these data for Figure 3? In subsection” iuGC treatment impairs the LDT-VTA circuitry”: Please check whether the p-value of 0.008 for the magnitude is correct. It seems improbable given that t(30) = 3.723.*

Thank you for this comment. We have rephrased the statistical methods used for the experiments. As stated above, we do not believe that we should compare both groups throughout development, since experiments were performed separately, so we have separated graphs according to age.

We do not understand the question regarding Figure 2, since this refers to an example of a waveform for a putative dopaminergic or GABAergic neuron in the VTA.

In addition, we have now included the statistical data for Figure 3.

We corrected the p-value in subsection “iuGC treatment impairs the LDT-VTA circuitry”: t(30)=3.723, p=0.0008. Thank you for finding this mistake.

*Reviewer #2:*

*[…] Figure 2 and 3. Specify what were the animals doing when the baseline described was acquired.*

Thank you for pointing this out. Electrophysiological recordings were acquired in anesthetized animals; we have included this description in the Results section of the manuscript (it was previously in the Materials and method section only).

*Present a distribution of the waveform duration vs. the firing rate.*

In accordance with the reviewer’s suggestion, we have now included additional graphs in Figure 2 and Figure 3 presenting the waveform duration vs. the firing rate of recorded cells.

*Discuss the mechanisms behind the difference in latencies when comparing the LDT-VTA responses en the control vs. the iuGC groups.*

Thank you for raising this point. So far, we have no data supporting any explanation for this observation. Our data suggests that both LDT and VTA neurons are changed by iuGC exposure (we see a reduction in TH staining in VTA for example – Leao et al., 2007). Both presynaptic (less acetylcholine being released for example?) and postsynaptic mechanisms (less receptors in VTA dendrites?) can contribute for the increased latency of response. However, because we do not have enough data to support either hypothesis, we decided not to discuss it in detail. However we added the following text (Discussion section): “Another remarkable finding was that the latency of excitatory responses in the VTA upon LDT electrical stimulation was substantially increased in iuGC animals, though so far we do not have any valid explanation for this. A combination of pre- and post-synaptic iuGC-induced changes may contribute for the observed increased latency in excitatory responses, thus more studies are needed in order to dissect this phenomenon.”

Does the reviewer have any ideas/suggestions?

*About the statement citing Figure 2: Altogether, these data demonstrated an imbalance in the excitatory and inhibitory inputs from the LDT. I suggest: Altogether, these data demonstrated an imbalance in the excitatory and inhibitory inputs to the VTA when electrically stimulating in the LDT.*

In accordance with the reviewer’s suggestion, we have corrected the above-mentioned sentence.

*Why did the optogenetic stimulation was not performed in the LDT?*

Neurons of the LDT are known to send axon collaterals to other brain regions that in turn could project to VTA, modulating VTA activity. To avoid this confounding/indirect effect, we have decided to activate only the terminals in the VTA. This was now explained in more detail in the manuscript in the Results subsection “iuGC treatment impairs the LDT-VTA circuitry”

*Please show images of the LDT cells expressing ChR2.*

We have included an image showing LDT cell soma expressing ChR2 (now Figure 2).

*Why, if Figure 3 shows overall more decreased responses, is this not reflected in Figure 3? May it be by a different probability to record from pDA or pGABAergic units in each group?*

We agree with the reviewer. In fact, the probability of recording pDAergic and pGABAergic units is different in the VTA region. We have a higher recruitment of pDAergic cells upon optical stimulation, which is in accordance with anatomical studies showing the preferential inputs of the LDT to mesoaccumbal DAergic cells in the VTA (Omelchenko and Sesack, 2005).

Figure 4.

*When presenting that activation of LDT terminals in the VTA during cue exposure period normalized the breakpoint of iuGC animals but had no effect in CTR animals, I suggest to place it next to the other protocol (both in text and figures). In this sense, it is clearer to the reader that LDT(ChR2)->VTA stimulation revert the effect on the break point rather than normalizing it.*

We are not certain that we understood this comment. We have made some changes in the behavioral graphs of the manuscript, please check.

Figure 4—figure supplement 1.

*For the analysis of locomotion in the LDT->VTA please plot the distance aligned to each time of the stimulation, that may reveal a different conclusion (see DOI: 10.1371/journal.pone.0033612).*

We analyzed the paper mentioned, but we did not fully understand what the reviewer asks for. We have now included an additional graph with total distance travelled. Please check if this is what you asked for.

[Editors' note: further revisions were requested prior to acceptance, as described below.]

*The reviewers have discussed the reviews with one another and the Reviewing Editor has drafted this decision to help you prepare a revised submission. Although the reviewers feel that the manuscript has been improved, they think some points need additional clarification before the study can be acceptable for publication.*

*Reviewer #2:*

*[…] Previously I requested a distribution of waveform duration versus the firing rate to be able to evaluate the criteria used for classification as pDAergic versus pGABAergic unit. Now when reviewing Figure 2 it is clear what criteria was used, however, it is not clear why there are few blue points (pDAergic) overlapping with the category "other" in Figure 3.*

The reviewer is completely right; there was a mistake in the categorization of 4 cells that should have been considered as “other” instead of DAergic. These cells presented a firing rate below 10Hz (firing rate of cell 1: 5.32; cell 2: 5.46; cell 3: 5.53; cell 4: 5.57) and waveform duration below 1.5 ms (duration of cell 1: 0.8; cell 2: 1.2; cell 3: 1.32; cell 4: 1.37). We have now reanalyzed all the electrophysiological data of Figure 2 and Figure 3 and corrected the figures and associated statistics. It is important to refer that reanalyzed results are very similar to previous ones and do not change the message of the manuscript.

*[…] Previously when requesting to discuss the mechanisms behind the difference in latencies when comparing the LDT-VTA responses in the control versus the iuGC groups I was looking for an explanation based on mechanism as now is provided (perhaps also discussing axonal conductivity, note that when stimulating the LDT axons in the VTA no changes in latency are observed; Figure —figure supplement 1), but also a discussion with the pre-existing measurements of this latency (if there is any by other groups), since a 1ms latency seems to fast, calling attention as if the stimulating electrode may not have been positioned exactly in the LDT.*

This is an interesting point. We have made an extensive revision of the bibliography and though many studies have performed electrical/optogenetic stimulation of LDT-VTA circuit, there is no information regarding the latency of responses.

Seminal studies (Forster et al., 2000; Yeomans, 2001, amongst others) performing electrical stimulation of the LDT and recordings in the VTA show electrophysiological responses in a widespread temporal window (minute scale), not presenting any data regarding latencies.

Recent studies (Lammel et al., 2012; Dautan et al., 2016; Steidl et al., 2017) using optogenetic manipulation of LDT-VTA circuit present behavioral and general electrophysiological responses to optical stimulation but no detailed information about types of cells that respond, latencies of responses etc. Therefore, the lack of information does not allow us to compare our responses to other studies in the field.

We would like to refer that, though we also think that the latency is very short, we do not believe that it is not due to incorrect positioning since we have confirmed the recording/stimulation electrode placement by histology for all animals (depicted in Figure 2).

Nevertheless, the reviewer raised an important point, which is the similar response latency upon optical stimulation of LDT-VTA terminals. Accordingly, we have briefly discussed this topic in the current version of the manuscript (Discussion section).

*[…] When asking to explain the differences in the previous Figure 3 (now 3E and 3G) and suggesting a possible explanation I am looking for a clearer explanation since it is not clear how to conceal the results of these two plots.*

We thank the reviewer for being persistent in this point. After reanalyzing the data, we found that the graph in Figure 3 had a mistake in the columns percentage. We have now re-confirmed all the electrophysiological data, corrected cell categorization, and calculated new percentages of response. A correct version of the graph is now provided, please see figure and associated legends and text (subsection “Activation of LDT terminals in the VTA rescues motivational deficits of iuGC animals”).

*[…] The point was that the optogenetic stimulation of the LDT terminals is not specific to normalize the breakpoint of iuGC animals, since activating this terminal using a stronger protocol also increase the break point in either control or iuGC animals.*

We have changed the word “normalize” to “revert” as the reviewer suggests – please see subsection “Surgery and cannula implantation”. We did not place the graphs with stronger optogenetic stimulation in main Figure 4 because this figure is already very busy with behavioral data.

*[…] When stating: Moreover, no effects in locomotion were observed using the same stimulation parameters in either group (Figure 4—figure supplement 1). This conclusion arises from the analysis performed (one point measured every tens of milliseconds) however an immediate effect on locomotion cannot be discharged (in the range of milliseconds). I previously pointed to the DOI: 10.1371/journal.pone.0033612 because it was one of the few papers at that time recognizing that there are motor modulations when manipulating VTA Dopaminergic cells activity. Please perform a smaller bin analysis of the data evaluating the manipulations on locomotion.*

Thank you for explaining this point in more detail. Unfortunately, we are not able to perform a smaller bin analysis of the locomotion data. The data output depends on the parameters chosen *a priori* during the behavior task, i.e. if data is selected to be collected at each second; we cannot analyze a smaller period than that.

Nevertheless, with the stimulation used in this study, we believe that we should not observe major effects in locomotion based on current bibliography: as elegantly showed in a study by Xiao and colleagues (Xiao et al., 2016), only when targeting the pedunculopontine nucleus, but not the LDT cholinergic projections specifically, they were able to see an effect on locomotor behavior. Additionally, in a previous study (Lammel et al., 2012), authors reported no effect of optical stimulation of all inputs from the LDT to the VTA in locomotion.